# STATE-SPACE MODELS CAN LEARN IN-CONTEXT BY GRADIENT DESCENT

## ABSTRACT

Deep state-space models (Deep SSMs) have shown capabilities for in-context learning on autoregressive tasks, similar to transformers. However, the architectural requirements and mechanisms enabling this in recurrent networks remain unclear. This study demonstrates that state-space model architectures can perform gradient-based learning and use it for in-context learning. We prove that a single structured state-space model layer, augmented with local sliding window attention, can reproduce the outputs of an implicit linear model with least squares loss after one step of gradient descent. Our key insight is that the diagonal linear recurrent layer can act as a gradient accumulator, which can be 'applied' to the parameters of the implicit regression model. We validate our construction by training randomly initialized augmented SSMs on simple linear regression tasks. The empirically optimized parameters match the theoretical ones, obtained analytically from the implicit model construction. Extensions to multi-step linear and non-linear regression yield consistent results. The constructed SSM encompasses features of modern deep state-space models, with the potential for scalable training and effectiveness even in general tasks. The theoretical construction elucidates the role of sliding window attention and multiplicative interactions in recurrent architectures as the key ingredients for enabling the expressive power typical of foundation models.

## 1 INTRODUCTION

The current generation of Large Language Models (LLMs) and foundation models are extremely capable and have started proliferating in several real-world applications. These models are based on the transformer architecture (Vaswani et al., 2017), and a big part of their capability has been attributed to in-context learning (Wei et al., 2023; Lu et al., 2023). In-context learning in transformers is relatively well studied (Wies et al., 2023; Pan et al., 2023; Guo et al., 2023; Garg et al., 2023). A prominent explanation for the mechanism used by transformers to do in-context learning is that the model performs in-context learning by gradient descent (von Oswald et al., 2023; Akyürek et al., 2024). But, the quadratic dependence of transformers on the input length makes them computationally expensive. To mitigate this, there has been much work on alternatives based on recurrent networks, such as state-space models (Gu et al., 2021; Gu & Dao, 2023) and linear recurrent networks (Orvieto et al., 2023). These recurrent models can perform inference efficiently since the computational complexity of recurrent networks is linear in the sequence length. At the same time, linear recurrent networks allow parallelization across the sequence during training using associative scan. The latest versions of these models are competitive with transformers at scale (Dao & Gu, 2024; De et al., 2024), and also capable of in-context learning (Grazzi et al., 2024). However, the mechanism they use for in-context learning remains unclear.

This work focuses on the in-context learning mechanism of State-Space Models (SSMs). Multiple variations of state-space models have recently shown competitive performance at scale (Gu & Dao, 2023; De et al., 2024), while earlier generations struggled with scaling (Smith et al., 2022; Poli et al., 2023). All SSMs, linear attention models, and other linear recurrent networks share a common formalism of being linear recurrent networks interleaved with non-linear layers. On the other hand, the ability to do in-context learning seems to be a hallmark of most recent scalable variants (Grazzi et al., 2024). Which features of these successful models contribute to in-context learning, as opposed

to earlier variants? Using a constructive approach, we pinpoint input-dependent input and output processing, as the key features required for in-context learning.

We show that SSMs with local sliding window attention, a form of input-dependent input processing, can perform in-context learning analogously to transformers, i.e. through gradient descent steps on an implicit linear regression problem. The key insight we use is that the state of the recurrent network can be used to aggregate gradients of the parameters of the implicit linear model, which can later be 'applied' to the initial parameters of the implicit model. Using the general SSM formalism, we show how to design the recurrent and output equations that enable them to do in-context learning. Our construction, which we call *GD-SSM*, results in a state-space model that is potentially applicable to general-purpose tasks as much as in-context learning tasks.

In summary, our contributions are to show that:

- one-layer SSMs with diagonal recurrence, two-dimensional state, and input-dependent input and output processing can perform one step of minibatch gradient descent on an implicit least-squares loss function;

- multi-step (minibatch) gradient descent can be achieved by stacking the 1-layer model;

- gradient descent for an implicit non-linear regression problem can be achieved by augmenting the SSMs with non-linearities;

- a randomly initialized model trained on regression tasks learns parameters that match our construction for in-context learning tasks based on regression.

## 2 BACKGROUND

A sequence model operates on an input sequence $\boldsymbol{S} = \{\boldsymbol{s}_t\}_{t=1}^T \in \mathbb{R}^{T \times f}$, where $T$ is also referred to as the context-length and $f$ is the feature dimension.

Contemporary sequence models based on transformers interleave self-attention with MLP layers to perform sequence processing. The self-attention performs sequence-mixing while the MLPs perform channel-mixing. The most common form of self-attention has been the scaled dot-product self-attention, which embeds the sequence into a query $\boldsymbol{Q} = \boldsymbol{S}\boldsymbol{W}_Q$, key $\boldsymbol{K} = \boldsymbol{S}\boldsymbol{W}_K$ and value $\boldsymbol{V} = \boldsymbol{S}\boldsymbol{W}_V$, where $\boldsymbol{W}_Q, \boldsymbol{W}_K \in \mathbb{R}^{f \times m}, \boldsymbol{W}_V \in \mathbb{R}^{f \times d}$, and then calculates the output of attention as

$$\mathrm{SA}(\boldsymbol{S}) = \mathrm{softmax}\left(\frac{\boldsymbol{Q}\boldsymbol{K}^T}{\sqrt{m}}\right)\boldsymbol{V}. \tag{1}$$

Sliding window attention (Beltagy et al., 2020) uses the same form as Eq. 1, but on an input sequence that is a subset of the full sequence, with a sliding window.

By discarding the softmax (and scaling for simplicity), self-attention can be written in a linear form

$$\mathrm{LSA}(\boldsymbol{S}) = \boldsymbol{Q}\boldsymbol{K}^T\boldsymbol{V}, \tag{2}$$

where LSA denotes linear self-attention.

Deep SSMs are sequence models that replace the self-attention with a linear recurrent network, to perform the sequence-mixing. In the most general form, the recurrent SSM block consists of a recurrent state $\boldsymbol{Z}_t \in \mathbb{R}^{d \times m}$ updated iteratively as

$$\boldsymbol{Z}_t = \boldsymbol{A}(\boldsymbol{s}_t) * \boldsymbol{Z}_{t-1} + \boldsymbol{B}(\boldsymbol{s}_t), \tag{3}$$

where $\boldsymbol{A}, \boldsymbol{B} : \mathbb{R}^f \to \mathbb{R}^{d \times m}$, and $*$ is some multiplication operator (e.g. matrix or element-wise multiplication).

The output of the SSM is often calculated by an input-dependent linear transformation of the current state

$$\boldsymbol{o}_t = \boldsymbol{Z}_t\boldsymbol{U}(\boldsymbol{s}_t), \tag{4}$$

combined with a non-linearity as

$$\boldsymbol{o}_t' = \boldsymbol{U}_2\left(\boldsymbol{o}_t \odot \sigma(\boldsymbol{U}_1\boldsymbol{o}_t)\right),$$

where $\odot$ is elementwise multiplication.

Transformers with linear attention can also be written in this recurrent form Katharopoulos et al. (2020a)

$$\boldsymbol{Z}_t = \boldsymbol{Z}_{t-1} + \boldsymbol{v}(\boldsymbol{s}_t) \otimes \boldsymbol{k}(\boldsymbol{s}_t)\,, \tag{5}$$

where $\boldsymbol{v}(\boldsymbol{s}_t) = \boldsymbol{W}_V^T \boldsymbol{s}_t \in \mathbb{R}^d$, $\boldsymbol{k}(\boldsymbol{s}_t) = \boldsymbol{W}_K^T \boldsymbol{s}_t \in \mathbb{R}^m$ and $\otimes$ is the outer product. This is a recurrent reformulation of the part of Eq. 2 containing $\boldsymbol{V}$ and $\boldsymbol{K}$ in recurrent form.

S4 (Gu et al., 2021) and multi-headed S5 (Smith et al., 2022) can also be written in this form

$$\boldsymbol{Z}_t = \boldsymbol{A}\boldsymbol{Z}_{t-1} + \boldsymbol{B}\boldsymbol{s}_t\,, \tag{6}$$

where $\boldsymbol{A}$ and $\boldsymbol{B}$ are learnable parameters of appropriate dimension, and $\boldsymbol{Z}_t$ consists of $m$ heads of dimension $d$ each (although multiple heads are not used in the original paper).

If sliding window attention is used to process the input before being fed into the SSM, all instances of $\boldsymbol{s}_t$ would be replaced by a context vector

$$\boldsymbol{c}_t = \mathrm{SA}(\boldsymbol{S}') \quad \text{or} \quad \boldsymbol{c}_t = \mathrm{LSA}(\boldsymbol{S}')\,,$$

where $\boldsymbol{S}' \subset \boldsymbol{S}$ is a subsequence of the whole sequence, and (L)SA denotes the (linear) self-attention operation.

## 3 SSMs CAN EMULATE GRADIENT DESCENT ON LINEAR REGRESSION TASKS

We will now show that an SSM as described in Section 2 can perform gradient descent on an implicit linear model to minimize a least squares loss (for particular choices of parameters). Extensions to non-linear regression models are considered in Section 3.3.

Consider a linear regression problem. The goal is to minimize the corresponding least squares loss using gradient descent. The linear model will be the implicit model on which the SSM performs gradient descent. Performing mini-batch (batch size > 1) gradient descent on the parameters of this implicit model involves two steps: (i) to accumulate gradients of the loss with respect to the parameters, and (ii) apply the accumulated gradient to the initial value of the parameters of the linear model to calculate the updated parameters. Predictions can be made with the updated parameters by combining them linearly with the input.

Assume the training samples for the linear regression problem are provided as a sequence of inputs and targets. A large enough SSM can then accumulate the gradients of the loss function in its state *if a sliding window attention-like layer processes the sequence inputs before the recurrence*[1].

Given the accumulated gradients, the next-step emission of the SSM is equivalent to i) updating the parameters of the implicit model gradient and ii) computing the model output with the updated parameters. Multiple steps of gradient descent can be achieved by stacking multiple layers, while nonlinearity in the implicit model can be handled by adding nonlinear input-output embedding layers. We argue that the architecture that allows a single layer to perform gradient descent provides the inductive bias for the model to do in-context learning.

### 3.1 SINGLE STEP 1-DIMENSIONAL LINEAR REGRESSION

Consider a linear regression model with 1-d output for simplicity

$$y = \boldsymbol{w}^T \boldsymbol{x}\,, \tag{7}$$

for parameter $\boldsymbol{w} \in \mathbb{R}^f$. This is the implicit linear model we aim to reproduce for in the 1-dimensional target case. Given dataset of $N$ samples $\mathcal{D} = \{\langle \boldsymbol{x}_i, y_i \rangle\}_{i=0}^N, \boldsymbol{x} \in \mathbb{R}^f, y \in \mathbb{R}$, the associated least squares loss is

$$\mathcal{L}(\mathcal{D}; \boldsymbol{w}) = \frac{1}{2N} \sum_{i=1}^N ||\hat{y}_i - y_i||_2^2 = \frac{1}{2N} \sum_i \left(\boldsymbol{w}^T \boldsymbol{x}_i - y_i\right)^2\,. \tag{8}$$

---

[1]A SSM layer with input-dependent recurrence would be able to simulate a sliding window attention layer, but with significantly increased computational and conceptual complexity.

The best fit for $\boldsymbol{w}$ is the minimum of $\mathcal{L}$ over $\boldsymbol{w} \in \mathbb{R}^f$. The gradient of the loss calculated on the first $t$ samples of the dataset is

$$\nabla_{\boldsymbol{w}} \mathcal{L}(\mathcal{D}_{1:t}; \boldsymbol{w}_0) = \frac{1}{t} \sum_{i=1}^{t} \left( \boldsymbol{w}_0^T \boldsymbol{x}_i - y_i \right) \boldsymbol{x}_i \,,$$

where $\mathcal{D}_{1:t}$ denotes the first $t$ samples in $\mathcal{D}$. The unscaled gradient,

$$\boldsymbol{g}_{\boldsymbol{w}_0}(\mathcal{D}_{1:t}) = \sum_{i=1}^{t} \left( \boldsymbol{w}_0^T \boldsymbol{x}_i - y_i \right) \boldsymbol{x}_i \,,$$

can be recursively calculated as

$$\boldsymbol{g}_{\boldsymbol{w}_0}(\mathcal{D}_{1:t}) = \boldsymbol{g}_{\boldsymbol{w}_0}(\mathcal{D}_{1:t-1}) + \left( \boldsymbol{w}_0^T \boldsymbol{x}_t - y_t \right) \boldsymbol{x}_t \,. \tag{9}$$

Scaling the $\boldsymbol{g}_{\boldsymbol{w}_0}(\mathcal{D}_{1:t})$ gives the mini-batch gradient $\nabla_{\boldsymbol{w}} \mathcal{L}(\mathcal{D}_{1:t}; \boldsymbol{w}_0) = \frac{1}{t} g_{\boldsymbol{w}_0}(\mathcal{D}_{1:t})$ for minibatch size $t$.

To make a prediction, $\hat{y}$, we apply the gradient to the parameters of the linear model in Eq. 7 and compute the corresponding output, i.e.

$$\hat{y}_{t+1} = \left( \boldsymbol{w}_0 - \eta \nabla_{\boldsymbol{w}_0} \mathcal{L}(\mathcal{D}_{1:t}; \boldsymbol{w}_0) \right)^T \boldsymbol{x}_{t+1} \,,$$
$$= \left( \boldsymbol{w}_0 - \frac{\eta}{t} g_{\boldsymbol{w}_0}(\mathcal{D}_{1:t}) \right)^T \boldsymbol{x}_{t+1} \,.$$

When $\boldsymbol{w}_0 = \boldsymbol{0}$, this reduces to

$$\hat{y}_{t+1} = -\frac{\eta}{t} \boldsymbol{g}_{\boldsymbol{w}_0}(\mathcal{D}_{1:t})^T \boldsymbol{x}_{t+1} \,. \tag{10}$$

**Implementation as an SSM:** Equation 9, which is a linear recurrence equation, can be implemented by an appropriately constructed SSM.

**Proposition 1** *Given a diagonal linear recurrent layer, and tokens $\boldsymbol{s}_j = \boldsymbol{c}_j = [\boldsymbol{x}_j y_j, \boldsymbol{x}_{j+1}]$, for $j = 1, \ldots, N$, and $[\ldots]$ concatenation, $\boldsymbol{x}_j, y_j$ drawn from a linear model, one can construct recurrent matrix $\boldsymbol{A}(\boldsymbol{s}_j)$, input $\boldsymbol{B}(\boldsymbol{s}_j)$ and output matrix $\boldsymbol{U}(\boldsymbol{s}_j)$ such that each recurrent step for every token $\boldsymbol{s}_j$ produces $\hat{y}_{j+1} = -(\Delta \boldsymbol{w})^T \boldsymbol{x}_{j+1}$ as output, where $\Delta \boldsymbol{w}$ is one step of gradient descent, i.e. $\Delta \boldsymbol{w} = \eta \nabla_{\boldsymbol{w}} \mathcal{L}$. The test input $\boldsymbol{x}_{N+1}$ is contained in token $\boldsymbol{c}_N$, and produces the test prediction $\hat{y}_{N+1}$.*

Specifically, if we use $\boldsymbol{z}_t \in \mathbb{R}^f$ to denote the state of the recurrent network and let it directly correspond to the vector $\boldsymbol{g}_{\boldsymbol{w}_0}(\mathcal{D}_{1:t})$, the equivalent SSM layer is a linear recurrence equation,

$$\boldsymbol{z}_t = \boldsymbol{I} \, \boldsymbol{z}_{t-1} + \left( \boldsymbol{w}_0^T \boldsymbol{x}_t - y_t \right) \boldsymbol{x}_t \,. \tag{11}$$

The state of the SSM, $\boldsymbol{z}_t$, represents the implicit linear regression problem through the *unscaled accumulated gradients* of the least squares loss with respect to the parameters $\boldsymbol{w}_0$ of the implicit linear model.

As linear regression is performed on the training dataset $\mathcal{D} = \{\langle \boldsymbol{x}_i, y_i \rangle\}_{i=0}^N$, the SSM receives the training data in the form of a sequence as input. In the most general case, this is a sequence $\boldsymbol{s}_1 = \boldsymbol{x}_1, \boldsymbol{s}_2 = [0, ..., y_1], \ldots$ where $[\ldots]$ denoting concatenation and $y_i$ is padded with $f - 1$ zeros for its dimensions to match that of $\boldsymbol{x}_i$. This more general case is discussed in the next section. But here, we will consider a case which simplifies our construction.

Let $\boldsymbol{s}_1, \boldsymbol{s}_2, \ldots$ be a sequence of constructed context vectors $\boldsymbol{s}_t = \boldsymbol{c}_t$, where each $\boldsymbol{c}_t = [\boldsymbol{x}_t y_t, \boldsymbol{x}_{t+1}] \in \mathbb{R}^{2f}$, and let us assume $\boldsymbol{w}_0 = \boldsymbol{0}$ for simplicity [2]. If the sequence input weights $\boldsymbol{\Psi} \in \mathbb{R}^{2f \times f}$ are such that $\boldsymbol{\Psi}^T \boldsymbol{c}_t = \boldsymbol{x}_t y_t$, Eq. 11 can be written as an SSM (Eq. 3), i.e.

$$\boldsymbol{z}_t = \boldsymbol{I} \, \boldsymbol{z}_{t-1} + \boldsymbol{\Psi} \, \boldsymbol{c}_t \,. \tag{12}$$

---

[2] The more general case is treated for the case of multi-step GD in Appendix A.2.

A parameter matrix, $\mathbf{\Psi}$, satisfies equation 12 if all but the first $f$ diagonal entries are zero. The state of this network is the unscaled gradient $t\,\nabla_{\boldsymbol{w}_0}\mathcal{L}(\mathcal{D}_{1:t};\boldsymbol{w}_0)$ and the state recursion accumulates the gradients as in step (i) above.

The accumulated gradient is then 'applied' to the implicit model's initial parameters, $\boldsymbol{w}_0$, before computing the $(N+1)$-th output. With $\boldsymbol{w}_0 = \boldsymbol{0}$, the output is

$$o_t = \beta \boldsymbol{z}_t^T \mathbf{\Theta} \boldsymbol{c}_t \,, \tag{13}$$

where $\beta = -\frac{\eta}{N}$, $\eta$ is the learning rate, and $N$ is the number of training points or, equivalently, the total length of the context. The SSM final output, $o_t$ above, corresponds to a prediction of the trained linear model, $\hat{y}_{t+1}$ in Eq. 10, if $\mathbf{\Theta}$ obeys $\mathbf{\Theta}\boldsymbol{c}_t = \boldsymbol{x}_{t+1}$. It is easy to check that $\mathbf{\Theta}$ satisfies this condition if all but its last $f$ diagonal entries are zero (see Figure 3 for a concrete example). Note that the output in Eq. 13 matches the general form of the SSM output in Eq. 4 (without the non-linearity).

The above shows that the following SSM

$$\boldsymbol{c}_t = [\boldsymbol{x}_t y_t, \boldsymbol{x}_{t+1}] \,, \tag{14}$$
$$\boldsymbol{z}_t = \boldsymbol{I}\,\boldsymbol{z}_{t-1} + \mathbf{\Psi}\,\boldsymbol{c}_t \,, \tag{15}$$
$$o_t = \beta \boldsymbol{z}_t^T \mathbf{\Theta} \boldsymbol{c}_t = \hat{y}_{t+1} \,. \tag{16}$$

can perform gradient descent on the parameters $\boldsymbol{w}_0$ of the implicit linear model and use this mechanism for in-context learning. The specific structure of the SSM in equation 14 demonstrates the importance of multiplicative processing, for both the inputs and outputs.

### 3.2 SINGLE STEP N-DIMENSIONAL LINEAR REGRESSION

In this section, we generalize the construction above to the N-dimensional case. Without loss of generality, we assume the input and the target, $\boldsymbol{x}$ and $\boldsymbol{y}$, have both dimensions $f$ i.e. $\boldsymbol{x}, \boldsymbol{y} \in \mathbb{R}^f$. If this is not the case, the input and output dimensionality can be matched by defining appropriate embeddings[3]. We can then treat the N-dimensional system as $f$ 1-D linear regression problems, one for each element of $\boldsymbol{y}$.

**Proposition 2** *Given a diagonal linear recurrent layer augmented with local sliding window attention with sliding window of size 3, and tokens $\boldsymbol{s}_{2j} = \boldsymbol{x}_j$ and $\boldsymbol{s}_{2j+1} = \boldsymbol{y}_j$, for $j = 1, \ldots, N$, $\boldsymbol{x}_j, \boldsymbol{y}_j$ drawn from a linear model, one can construct recurrent matrix $\boldsymbol{A}(\boldsymbol{s}_j)$, input $\boldsymbol{B}(\boldsymbol{s}_j)$ and output matrix $\boldsymbol{U}(\boldsymbol{s}_j)$ such that each recurrent step for every token $\boldsymbol{s}_j$ produces $\hat{\boldsymbol{y}}_{j+1} = -(\Delta \boldsymbol{W})^T \boldsymbol{x}_{j+1}$ as output, where $\Delta \boldsymbol{W}$ is one step of gradient descent, i.e. $\Delta \boldsymbol{W} = \eta \nabla_{\boldsymbol{W}}\mathcal{L}$. The test input $\boldsymbol{x}_{N+1}$ is contained in token $\boldsymbol{s}_{2N+2}$, and produces the test prediction $\hat{\boldsymbol{y}}_{N+1}$.*

Similar to Eq. 11, we show the above by writing the SSM as

$$\boldsymbol{Z}_t = \boldsymbol{Z}_{t-1} + \boldsymbol{y}_t\,\boldsymbol{x}_t^T \,, \tag{17}$$

where $\boldsymbol{Z}_t$ corresponds to the parameters of the implicit linear model, $\boldsymbol{W} \in \mathbb{R}^{f \times f}$, and we assume, for simplicity, that $\boldsymbol{W} = \boldsymbol{0}$[4]. The output is

$$\boldsymbol{o}_t = \beta \boldsymbol{Z}_t\,\boldsymbol{x}_{t+1} \,. \tag{18}$$

See Appendix A.1 for the full derivation.

To see how this can be written in the form of Eq. 3, let the input sequence consist of the training dataset of the implicit linear regression problem (as before). This time, we cast the training dataset into a standard sequence $\boldsymbol{s}_1, \boldsymbol{s}_2, \ldots$, where

$$\boldsymbol{s}_{2j} = \boldsymbol{x}_j \,, \tag{19}$$
$$\boldsymbol{s}_{2j+1} = \boldsymbol{y}_j \,, \tag{20}$$

---

[3]E.g. with dimensions $k, l$ into the same higher dimension $k + l = f$ by concatenation with appropriately sized zero vectors, or through a linear transformation to dimension $f$.

[4]The more general case is discussed in Appendix A.2.

and $\boldsymbol{s}_i$ denotes the $i$-th sequence element of the input, such that $\boldsymbol{s}_{2j+2} = \boldsymbol{x}_{j+1}$.

At each step, the state update, in Eq. 17, and the output, in Eq. 18, include $\boldsymbol{x}_t, \boldsymbol{y}_t, \boldsymbol{x}_{t+1}$. This necessitates the introduction of a quadratic (in $\mathbf{s}_i$) sliding window attention mechanism. Let $\boldsymbol{C}_t$ be the context matrix for the sliding window attention, i.e. a sliding window of length three running through the sequence,

$$\boldsymbol{C}_t = \begin{bmatrix} \vdots & \vdots & \vdots \\ \boldsymbol{x}_t & \boldsymbol{y}_t & \boldsymbol{x}_{t+1} \\ \vdots & \vdots & \vdots \end{bmatrix} . \tag{21}$$

The sliding window attention operation, $\boldsymbol{C}\boldsymbol{Q}\boldsymbol{C}^T$, is a truncated form of Eq. 2. When $\boldsymbol{Q} = \begin{pmatrix} 1 \\ 0 \\ 0 \end{pmatrix} \begin{pmatrix} 0 & 1 & 0 \end{pmatrix} = \begin{bmatrix} 0 & 1 & 0 \\ 0 & 0 & 0 \\ 0 & 0 & 0 \end{bmatrix}$, plugging $\boldsymbol{C}\boldsymbol{Q}\boldsymbol{C}^T$ into the second term in Eq. 17 produces an SSM with input $\boldsymbol{y}_t \boldsymbol{x}_t^T$.

The SSM, which corresponds to GD-SSM, can now be written as in Eq. 3, i.e.

$$\boldsymbol{Z}_t = \boldsymbol{Z}_{t-1} + \boldsymbol{C}_t \boldsymbol{Q} \boldsymbol{C}_t^T ,$$

with output

$$\boldsymbol{o}_t = \beta \boldsymbol{Z}_t \boldsymbol{C}_t \boldsymbol{q} ,$$

where $\boldsymbol{q} = \begin{pmatrix} 0 \\ 0 \\ 1 \end{pmatrix}$ makes the output $\hat{\boldsymbol{y}}_{t+1}$ as in Eq. 10. The output also matches the form of the general SSM output in Eq. 4. The construction allows us to perform gradient descent on the parameters of an arbitrary dimensional implicit model. We call this type of SSM a GD-SSM. When we train the model for ICL tasks, the parameters $\boldsymbol{Q}, \boldsymbol{q}$ as well as the embeddings for $\boldsymbol{x}, \boldsymbol{y}$ are randomly initialized and trained using a mean-squared error loss.

Eq. 3 shows that the recurrent state of the SSM is 2-dimensional, as in most recently proposed SSMs (Dao & Gu, 2024). The sliding window attention reproduces the local linear self-attention of Eqs. 2 and 5. As for the 1-dimensional case, the self-attention higher-order dependencies, $\boldsymbol{C}\boldsymbol{Q}\boldsymbol{C}^T$, is key for making the SSM learn (in-context) an implicit (linear) regression model.

### 3.3 GENERALISING TO ANY REGRESSION PROBLEM

**Multi-step gradient descent:** The proposed construction can be extended to multi-step gradient descent. Since each layer of a GD-SSM produces the parameters of the implicit linear model updated by one step of GD, this is equivalent to stacking together multiple layers. In our derivations above, we assumed the initial parameter of the implicit linear model is $\boldsymbol{0}$. In the multi-step GD, all layers other than the first will correspond to a *non-zero initialised implicit linear model*. Technically, extra gradient steps in the implicit model introduce one additional term in the gradient accumulation equation. Each of the two terms requires a separate (parallel) recurrence and is performed by a dedicated layer. At the end, the states are combined to obtain the multiple-step GD update, with minor extra computational burdens. See Appendix A.2 for a detailed construction of multi-step GD.

**Non-linear regression:** Non-linear regression can be handled by adding MLP layers to the GD-SSM. In the previous sections, we let GD-SSM accumulate the gradients of a linear regressor. Additional MLP layers can learn to transform the state of these linear layers into quantities corresponding to the gradient of the implicit non-linear model. See Appendix A.3 for a detailed explanation.

**Regularisation terms in the loss:** As the recurrent layers only accumulate the gradients, we can separate the gradient calculation and accumulation from its application. This has a practical advantage. Any input-independent regularisation term, e.g. L2 norms, can be added to the model without changing the recurrence structure. See Appendix A.4 for details.

# 4 TRAINED LINEAR RECURRENT NETWORKS DO EMULATE GRADIENT DESCENT ON LINEAR REGRESSION TASKS

We investigated if the GD-SSM variant of the general SSM architecture does do gradient descent in-context learning. To do this, we trained a randomly initialised model on various in-context learning tasks for linear and non-linear regression. In each case, the sequence token input to the model consisted of the inputs $\boldsymbol{x}$ and target values $\boldsymbol{y}$ from the training dataset $\mathcal{D}$, and the model was expected to output the prediction for the query (test) $\boldsymbol{x}$ given in the last timestep. The models were trained to minimize the mean squared error between the test prediction and target.

## 4.1 SINGLE STEP 1-DIMENSIONAL LINEAR REGRESSION

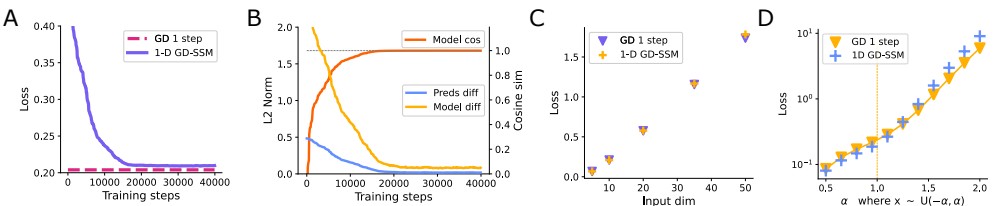

Figure 1: **Comparing one step of GD with a trained single layer GD-SSM for 1-dimensional regression**: **A**: Trained single layer GD-SSM loss and GD-SSM loss with the parameters from our construction are identical. **B**: Cosine similarity and the L2 distance between models as well as their predictions. **C**: Comparison of loss between Gradient Descent (GD) and the SSM layer model for different input sizes $N = N_x$. **D**: The trained single 1-D SSM layer, and gradient descent show identically loss (in log-scale) when provided input data different than during training i.e. with scale of 1. We display the mean/std. or the single runs of 5 seeds.

We first tested the simplest case of one-step gradient descent on a linear regression problem with scalar predictions/targets. This corresponds directly to the construction in Section 3.1. To do this, similar to Garg et al. (2023); von Oswald et al. (2023), we randomly generated linear regression tasks consisting of training and test points, and trained the model to make a prediction for the test input using the training input as context.

We generated randomly sampled linear regression tasks $\tau$ in the following way: Each task (context) $\tau$ consisted of a sequence of in-context training data $\mathcal{D}_\tau = \{\langle \boldsymbol{x}_{\tau,i}, y_{\tau,i} \rangle\}_{i=1}^{N}$ and test point $\langle \boldsymbol{x}_{\tau,N+1}, y_{\tau,N+1} \rangle$. To generate this, the $\boldsymbol{x}_{\tau,i}$s are sampled from a uniform distribution $\boldsymbol{x}_{\tau,i} \sim U(-1,1)^f$. Then, for each task $\tau$, the parameters of its implicit linear model $\boldsymbol{w}_\tau$ is sampled from a normal distribution, so that each element $[\boldsymbol{w}_\tau]_i \sim \mathcal{N}(0,1)$. This is used to calculate the $y_{\tau,i}$s for each corresponding $\boldsymbol{x}_{\tau,i}$ using $y_{\tau,i} = \boldsymbol{w}_\tau^T \boldsymbol{x}_{\tau,i}$.

The sequence $\boldsymbol{S} = \{\boldsymbol{s}_{\tau,1}, ..., \boldsymbol{s}_{\tau,N}\}$ is constructed so that $\boldsymbol{s}_{\tau,t} = \boldsymbol{c}_{\tau,t} = [\boldsymbol{x}_{\tau,t} y_{\tau,t}, \boldsymbol{x}_{\tau,t+1}]$, with $[\ldots]$ denoting the vector concatenation operation, and $\boldsymbol{c}_t$ is the constructed context vector. Note that this includes the query $\boldsymbol{x}_{\tau,N+1}$ in $\boldsymbol{c}_N$. We will use a more general construction in the next section. The outputs of the GD-SSM at time $T = N$ is the prediction for $\boldsymbol{x}_{N+1}$ i.e. $SSM(\boldsymbol{S}_\tau)_N = \hat{y}_{\tau,N+1}$, with target $y_{\tau,N+1} = \boldsymbol{w}_\tau \boldsymbol{x}_{\tau,N+1}$. The model was trained to minimize the expected squared prediction error, averaged over linear regression tasks $\tau$:

$$\min_\theta \mathbb{E}_\tau \left[ \left\| \hat{y}_\theta \left( \boldsymbol{c}_{\tau,1}, \ldots, \boldsymbol{c}_{\tau,N} \right) - y_{\tau,\text{test}} \right\|^2 \right],$$

where $\theta$ are the randomly initialised parameters of the GD-SSM. We evaluate our model on multiple metrics:

1. L2 norm between the difference in predictions $\|\hat{y}_\theta(x_{\tau,\text{test}}) - \hat{y}_{\theta_\text{GD}}(x_{\tau,\text{test}})\|_2$ where $\hat{y}_{\theta_\text{GD}}$ is the prediction from the GD based construction.

2. Cosine similarity between the sensitivities $\frac{\partial \hat{y}_{\theta_\text{GD}}(x_{\tau,\text{test}})}{\partial x_\text{test}}$ and $\frac{\partial \hat{y}_\theta(x_{\tau,\text{test}})}{\partial x_\text{test}}$.

3. L2 norm between the sensitivites.

4. Loss of the two models where validation data is sampled from $U(-\alpha, \alpha)^{n_i}$, with a different $\alpha$ than the one used in training.

5. The loss of two models when trained for different number of feature dimensions $f$.

Figure 1 shows the results of this comparisons. We find that these metrics show excellent agreement between the trained model and GD, over different hyperparameters. To test if the trained network has learned a general purpose learning rule as opposed to fitting to the dataset, that is to test if the trained network generalises to linear regression tasks outside the training distribution, we draw values of inputs from $U(-\alpha, \alpha)$. We see that our model generalises to both cases.

### 4.2 SINGLE STEP N-DIMENSIONAL LINEAR REGRESSION

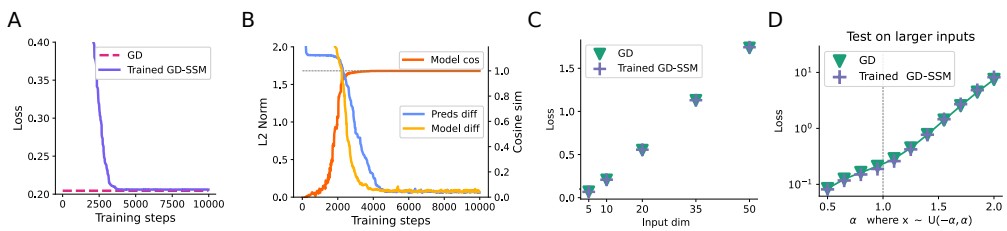

Figure 2: **Comparing one step of GD with a trained single layer GD-SSM for N-dimensional regression**: **A**: Trained single layer GD-SSM loss and GD-SSM loss with the parameters from our construction are identical. **B**: Cosine similarity and the L2 distance between models as well as their predictions. **C**: Comparison of loss between Gradient Descent (GD) and the SSM layer model for different input sizes $f$. **D**: The trained GD-SSM layer, and gradient descent show identically loss (in log-scale) when provided input data different than during training i.e. with scale of 1. We display the mean/std. or the single runs of 5 seeds.

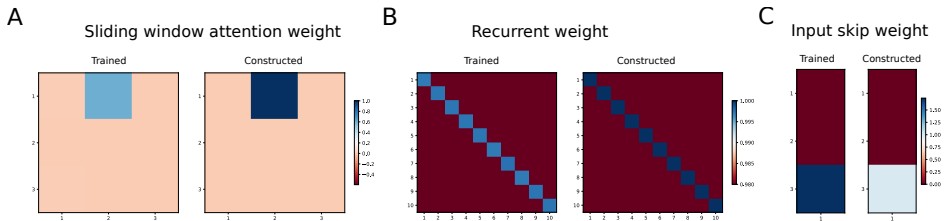

Figure 3: Comparison of learned weights a GD-SSM with the GD-SSM parameters from our construction. **A:** Comparison of local self attention weights of trained GD-SSM with the GD-SSM parameters from our construction. **B:** Comparison of recurrent parameters of trained GD-SSM with the GD-SSM parameters from our construction. Since the recurrence parameters are tensors, for the ease of visualisation, each diagonal entry is the mean of the corresponding diagonal recurrence matrix. **C:** Comparison of skip connection weights of trained GD-SSM with the GD-SSM parameters from our construction.

In the more general case where we allow the targets $\boldsymbol{y}_{\tau,i}$ to be of arbitrary dimension, we also relax other assumptions in the form of the input. Notably, instead of constructing the inputs in a particular way, we let the sequence be similar to a more natural sequence of $\boldsymbol{x}_1, \boldsymbol{y}_1, \boldsymbol{x}_2, \boldsymbol{y}_2, \ldots$, i.e. $\boldsymbol{s}_{2j} = \boldsymbol{x}_j; \boldsymbol{s}2j + 1 = \boldsymbol{y}_j$. This requires the introduction of a sliding window attention mechanism with attention window of 3 as discussed in Section 3.2. The sliding window attention mechanism

allows the model to access inputs from multiple adjacent timesteps, and not just the current one. We perform the same comparisons as in Section 4.1, and find that the model trained from a random initialisation has an excellent match with the construction, both in the output/loss metrics (Figure 2) and in the parameters it ends up with (Figure 3).

We compared loss metrics for GD-SSM with other standard models including one- and two-layer transformers, and S5 (Smith et al., 2022). We found that, on exactly the same sequence token inputs, only the GD-SSM and a 2-layer transformer reached the same loss as actual GD, while other SSMs such as S5 struggle to perform ICL with 1 layer (Figure 4C).

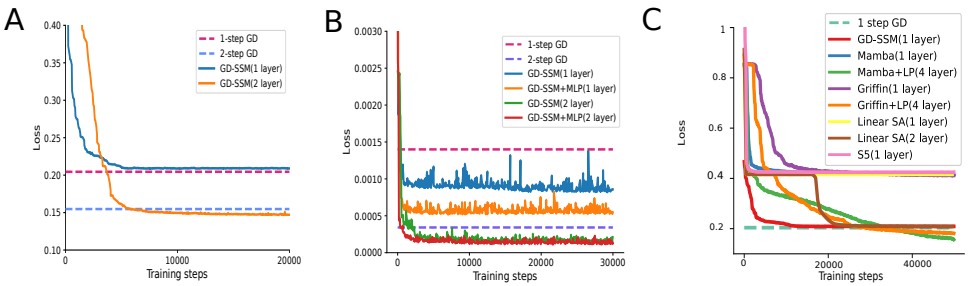

Figure 4: **A**: Comparison of GD-SSM performance with single layer and two layers on regression task. **B**: Comparison of GD-SSM performance with and without MLP layers in single layer and multi-layer setup on non-linear regression task. **C**: Comparison with other models on 1-D regression. The GD-SSM model was evaluated with both 1-layer and 2-layer configurations, and the S5, Mamba, and Griffin models were included for comparison. TF refers to linear Transformer models, with both 1-layer and 2-layer variants tested to evaluate their performance. A similar plot for N-D regression is shown in Figure 6.

### 4.3 Multi-step and non-linear regression

To perform multi-step GD, we tested a GD-SSM with multiple-layers, where each layer of the GD-SSM does 1-step of GD as shown in Appendix A.2. The token construction is identical to that used for N-dimensional linear regression, and each layer includes the sliding window attention mechanism. In Figure 4A, we show that that trained network is able to reach the same loss as the multi-step GD.

Finally, to be able to handle any regression task, we tested GD-SSM with an MLP layer on non-linear regression tasks. Again, we see that the trained model is able to reach the same level of performance as performing GD directly on the dataset (Figure 4B).

## 5 Related work

In-context learning in transformers has been studied extensively, and various mechanisms have been proposed to explain it (Hendel et al., 2023; von Oswald et al., 2023; Akyürek et al., 2022). Of those, the most prominent is that the self-attention mechanism performs gradient descent on a linear loss. A construction with linear self-attention was demonstrated by von Oswald et al. (2023). While linear-self attention can be written as an RNN (Katharopoulos et al., 2020b), the results of von Oswald et al. (2023) depends on the tokens being constructed in a specific way, which limits the generality of their construction. Moreover, the most common type of self-attention used is softmax scaled dot-product self-attention rather than linear self-attention, and the basic linear self-attention mechanism used in von Oswald et al. (2023) is not competitive with softmax self-attention. Whereas, we show a construction that uses local-self attention with state-space models, which has been shown to be competitive with softmax self-attention transformers (De et al., 2024). Zucchet et al. (2023) show a correspondence between gated linear models (which include many commonly used forms of SSMs) and linear self-attention, which is known to be able to do ICL. But compared to both von Oswald et al. (2023) and Zucchet et al. (2023), our model is more parsimonious because of

the following: (1) In general, a linear self-attention model requires $3f^2$ parameters ($f^2$ each for query, key and value) for input dimension $f$[5] and without including the output projection, whereas our model equivalently only requires $f^2$ parameters, one each for each of the $f^2$ recurrent units. (2) linear self-attention as constructed in von Oswald et al. (2023) requires $12f^2$ parameters to do GD on a $f \times f$ linear regression problem, whereas ours still only requires $f^2$ parameters. (3) to construct a gated linear recurrent model from linear self-attention as done in  Zucchet et al. (2023) requires $\mathcal{O}3(d^4)$ parameters for a linear self-attention with $3d^2$ parameters. This blows up the size of the model very significantly. On the other hand, since ours is a explicit and direct construction, our model size remain proportional to the size of the linear regression problem.

Liu et al. (2024) formulate the SSM layer as an online optimization objective with exact solution. Similarly, Sun et al. (2024) use the state of the SSM to perform GD. But both these papers consider online updates, that is minibatch size 1 updates. One layer of their model is not capable of performing larger minibatch updates. Moreover, without sliding window attention, their online optimization objective is limited to using single sequence tokens with a single-layer. And finally their goal is not to explain how in-context learning happens with existing architectures, but rather to develop entirely new SSM variants using the idea that an optimization process can be used for compressing information as well, which is the key property required of a recurrent network.

The combination of local-self attention with Mamba has been used in for improving performance in multiple instances, including most recently De et al. (2024); Ren et al. (2024). But these papers do not specifically construct their model for ICL, nor do they attempt to distill inductive biases required for ICL in state-space models.

## 6 DISCUSSION

This work establishes a clear connection between state-space models (SSMs) and gradient-based in-context learning. We have demonstrated, both theoretically and empirically, that SSMs can emulate gradient descent on implicit regression models, providing a mechanistic explanation for their in-context learning capabilities. Our construction, GD-SSM, reveals the crucial role of architectural features such as sliding window attention and multiplicative output interactions in enabling this behavior. These findings not only explain the success of recent SSM variants in in-context learning tasks but also provide valuable insights for the design of future sequence models.

The alignment between our theoretical construction and the behavior of trained models suggests that the gradient descent mechanism may be a natural inductive bias in these architectures. This understanding opens new avenues for analyzing and improving sequence models, potentially leading to more efficient and effective architectures for a wide range of tasks.

Future research could explore the implications of this mechanism in larger-scale models, more complex tasks, and real-world applications. Additionally, investigating how this understanding can be leveraged to enhance the design and training of state-space models could yield significant advancements in the field of sequence modeling.

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

# A   APPENDIX

## A.1   N-D LINEAR REGRESSION

The implicit linear model of N-dimensional ($N = f$) linear regression is

$$\hat{\boldsymbol{y}} = \boldsymbol{W}^T \boldsymbol{x},$$

for $\boldsymbol{W} \in \mathbb{R}^{f \times f}$ and $\hat{\boldsymbol{y}}, \boldsymbol{x} \in \mathbb{R}^f$. The loss is

$$\mathcal{L}(\mathcal{D}; \boldsymbol{W}_0) = \frac{1}{2N} \sum_i \left( \boldsymbol{W}_0^T \boldsymbol{x}_i - \boldsymbol{y}_i \right)^T \left( \boldsymbol{W}_0^T \boldsymbol{x}_i - \boldsymbol{y}_i \right). \tag{22}$$

The gradient of this loss calculated using the first $t$ samples $\mathcal{D}_{1:t}$ is

$$\nabla_{\boldsymbol{W}} \mathcal{L}(\mathcal{D}_{1:t}; \boldsymbol{W}_0) = \frac{1}{t} \sum_{i=1}^{t} \frac{\partial}{\partial W} \left( \boldsymbol{W}^T \boldsymbol{x}_i - \boldsymbol{y}_i \right)^T \left( \boldsymbol{W}^T \boldsymbol{x}_i - \boldsymbol{y}_i \right) \Big|_{\boldsymbol{W} = \boldsymbol{W}_0}.$$

For notational simplicity, we will write this as $f$ 1-D regression problems, one for each element of $\boldsymbol{y}$. Using $[\boldsymbol{W}]_{:,i}$ to denote the $i$-th column of matrix $\boldsymbol{W}$, using $[\boldsymbol{W}_0]_{:,i}$ to denote the $i$-th column of matrix $\boldsymbol{W}_0$ and using $[\boldsymbol{y}_t]_i = [\boldsymbol{W}]_{:,i}^T \boldsymbol{x}_t$ to denote the $i$-th element of vector $\boldsymbol{y}_t$,

$$\boldsymbol{g}(\mathcal{D}_{1:t}; [\boldsymbol{W}_0]_{:,i}) = \boldsymbol{g}(\mathcal{D}_{1:t-1}; [\boldsymbol{W}_0]_{:,i}) + \left( [\boldsymbol{W}_0]_{:,i}^T \boldsymbol{x}_t - [\boldsymbol{y}_t]_i \right) \boldsymbol{x}_t, \tag{23}$$

where $\frac{1}{t} \boldsymbol{g}(\mathcal{D}_{1:t}; [\boldsymbol{W}_0]_{:,i}) = [\nabla_{\boldsymbol{W}} \mathcal{L}(\mathcal{D}_{1:t}; \boldsymbol{W}_0)]_{:,i}$.

**Implementation as a linear recurrent network:**   The input is given as in Eqs. 19 & 20, with context matrix constructed as in Eq. 21.

The local self attention calculates $\boldsymbol{C}\boldsymbol{Q}\boldsymbol{C}^T$, which is then provided as input to the SSM layer. If $\boldsymbol{Q} = \begin{pmatrix} 1 \\ 0 \\ 0 \end{pmatrix} \begin{pmatrix} 0 \\ 1 \\ 0 \end{pmatrix}^T = \begin{bmatrix} 0 & 1 & 0 \\ 0 & 0 & 0 \\ 0 & 0 & 0 \end{bmatrix}$, this corresponds to the input to the SSM being $[\boldsymbol{y}_t]_i \boldsymbol{x}_t$ (assuming $\boldsymbol{W} = \boldsymbol{0}$).

Defining $\boldsymbol{Z}_t \in \mathbb{R}^{f \times f}$ as the state of the SSM, where $[\boldsymbol{Z}_t]_{:,i} = \boldsymbol{g}(\mathcal{D}_{1:t}; [\boldsymbol{W}_0]_{:,i})^T$ (note the transpose), and assuming $\boldsymbol{W}_0 = \boldsymbol{0}$, the equations Eq. 23 for all $i$s (all columns) can be written as a single equation

$$\boldsymbol{Z}_t = \boldsymbol{Z}_{t-1} + \boldsymbol{C}_t \boldsymbol{Q} \boldsymbol{C}_t^T,$$

with output

$$\boldsymbol{o}_t = -\frac{\eta}{N} \boldsymbol{Z}_t \boldsymbol{C}_t \boldsymbol{q}.$$

If $\boldsymbol{q} = \begin{pmatrix} 0 \\ 0 \\ 1 \end{pmatrix}$, this corresponds exactly to the output $\hat{\boldsymbol{y}}_{t+1} = -\eta \nabla_{\boldsymbol{W}} \mathcal{L}(\mathcal{D}_{1:t}; \boldsymbol{W}_0)^T \boldsymbol{x}_{t+1}$. This provides a construction that allows us to perform gradient descent on the parameters of an arbitrary dimensional input model.

## A.2   MULTI-STEP GD

**Proposition 3** *Given $1, \ldots, L$ diagonal linear recurrent layers, each augmented with sliding window attention with sliding window of size 3, and sequence tokens $\boldsymbol{s}_{2j} = \boldsymbol{x}_j$ and $\boldsymbol{s}_{2j+1} = \boldsymbol{y}_j$, for $j = 1, \ldots, N$, drawn from a linear model, for each layer $l$, one can construct pairs of recurrent matrices $\boldsymbol{A}^{(l,1)}(\boldsymbol{s}_j), \boldsymbol{A}^{(l,2)}(\boldsymbol{s}_j)$, inputs $\boldsymbol{B}^{(l,1)}(\boldsymbol{s}_j), \boldsymbol{B}^{(l,2)}(\boldsymbol{s}_j)$ and output matrices $\boldsymbol{U}^{(l,1)}(\boldsymbol{s}_j), \boldsymbol{U}^{(l,2)}(\boldsymbol{s}_j)$ such that each recurrent step for every token $\boldsymbol{s}_j$ produces $\hat{\boldsymbol{y}}_{j+1} = (\Delta_l \boldsymbol{W})^T \boldsymbol{x}_{j+1}$ as output, where $\Delta_l \boldsymbol{W}$ is the update for $l$ steps of gradient descent, i.e. $\Delta_l \boldsymbol{W} = ((\boldsymbol{W}_0 - \eta \nabla_{\boldsymbol{W}} \mathcal{L}(\mathcal{D}; \boldsymbol{W}_0)) - \eta \nabla_{\boldsymbol{W}} \mathcal{L}(\mathcal{D}; (\boldsymbol{W}_0 - \eta \nabla_{\boldsymbol{W}} \mathcal{L}(\mathcal{D}; \boldsymbol{W}_0))))...l$ times). The test input $\boldsymbol{x}_{N+1}$ is contained in token $\boldsymbol{s}_{2N+2}$, and produces the test prediction $\hat{\boldsymbol{y}}_{N+1}$.*

We have so far assumed that $\boldsymbol{W}_0 = \boldsymbol{0}$ thus far, which for the first layer corresponds to initialising the parameters of the equivalent linear model to all zeros. For multi-step GD, the second layer onwards have a non-zero initial value of parameters, so let's derive a general form for a layer that performs GD with non-zero initialisation.

Starting from Eq. 23, repeated below for convenience,

$$\boldsymbol{g}(\mathcal{D}_{1:t}; [\boldsymbol{W}_0]_{:,i}) = \boldsymbol{g}(\mathcal{D}_{1:t-1}; [\boldsymbol{W}_0]_{:,i}) + \left([\boldsymbol{W}_0]_{:,i}^T \boldsymbol{x}_t - [\boldsymbol{y}_t]_i\right) \boldsymbol{x}_t,$$

we note that this accumulates two different components – $\left([\boldsymbol{W}_0]_{:,i}^T \boldsymbol{x}_t\right) \boldsymbol{x}_t$ and $[\boldsymbol{y}_t]_i \boldsymbol{x}_t$.

We propose having two different heads (per layer) to accumulate these quantities separately (i) $[\boldsymbol{y}_t]_i \boldsymbol{x}_t$ and (ii) $\boldsymbol{x}_t \boldsymbol{x}_t$.

Defining $\boldsymbol{Z}_t \in \mathbb{R}^{m \times m}$ as the state of the SSM, where $[\boldsymbol{Z}_t]_{:,i} = \boldsymbol{g}(\mathcal{D}_{1:t}; [\boldsymbol{W}_0]_{:,i})^T$ (note the transpose), the recurrent network corresponding to the accumulation of (i) is the same as before:

$$\boldsymbol{Z}_t = \boldsymbol{Z}_{t-1} + \boldsymbol{C}_t \boldsymbol{Q} \boldsymbol{C}_t^T.$$

For (ii), we can write an equivalent recurrent network

$$\tilde{\boldsymbol{Z}}_t = \tilde{\boldsymbol{Z}}_{t-1} + \boldsymbol{C}_t \tilde{\boldsymbol{Q}} \boldsymbol{C}_t^T,$$

If $\tilde{\boldsymbol{Q}} = \begin{pmatrix} 1 \\ 0 \\ 0 \end{pmatrix} \begin{pmatrix} 1 \\ 0 \\ 0 \end{pmatrix}^T = \begin{bmatrix} 1 & 0 & 0 \\ 0 & 0 & 0 \\ 0 & 0 & 0 \end{bmatrix}$, this corresponds to the input to the SSM being $\boldsymbol{x}_t \boldsymbol{x}_t^T$.

The output at layer $l$ is

$$\boldsymbol{o}_t^{(l)} = (\boldsymbol{W}_{l-1} + \Delta_l \boldsymbol{W}) \, \boldsymbol{C}_t \, \boldsymbol{q},$$

where

$$\Delta_l \boldsymbol{W} = -\frac{\eta}{N} \left(\boldsymbol{W}_{l-1}^T \tilde{\boldsymbol{Z}}_t^{(l)} - \boldsymbol{Z}_t^{(l)}\right).$$

In summary, at each layer, there are two linear recurrent layers, and the output includes a multiplicative combination across layers and with the external input.

Since each recurrent layer is performing the same operation, one could also loop the output of each layer back to do multiple steps of GD.

### A.3  NON-LINEAR GD

Let us consider a non-linear regression problem with a least squares loss, where the output labels are again 1-dimensional for simplicity. For a given dataset of $N$ samples $\mathcal{D} = \{<\boldsymbol{x}_i, y_i>\}_{i=0}^N, \boldsymbol{x} \in \mathbb{R}^f, y \in \mathbb{R}$, predictions from a non-linear model are generated using

$$\hat{y} = g(\boldsymbol{w}^T \boldsymbol{x}),$$

for $\boldsymbol{w} \in \mathbb{R}^f$ and where $g$ is some non-linear function such as sigmoid or an MLP. This will be our implicit non-linear model for this 1-dimensional target case.

The best fit for $\boldsymbol{w}$ is found by minimizing the loss

$$\mathcal{L}(\mathcal{D}; \boldsymbol{w}_0) = \frac{1}{2N} \sum_{i=1}^N ||\hat{y}_i - y_i||_2^2 = \frac{1}{2N} \sum_i \left(g(\boldsymbol{w}_0^T \boldsymbol{x}_i) - y_i\right)^2.$$

The gradient of the loss calculated on the first $t$ samples of the dataset is

$$\nabla_{\boldsymbol{w}} \mathcal{L}(\mathcal{D}_{1:t}; \boldsymbol{w}_0) = \frac{1}{t} \sum_{i=1}^t \left(g(\boldsymbol{w}_0^T \boldsymbol{x}_i) - y_i\right) g'(\boldsymbol{w}_0^T \boldsymbol{x}_i) \boldsymbol{x}_i,$$

where $\mathcal{D}_{1:t}$ denotes the first $t$ samples in $\mathcal{D}$, and $g'$ denotes the first derivative of $g$.

We define the term $\boldsymbol{g}$ but without containing $g$, as

$$\boldsymbol{g}_{\boldsymbol{w}_0}(\mathcal{D}_{1:t}) = \sum_{i=1}^t \left(\boldsymbol{w}_0^T \boldsymbol{x}_i - y_i\right) \boldsymbol{x}_i,$$

which can be recursively calculated as before. Note that this quantity is not the unscaled gradient anymore. In this case we have the following Proposition:

**Proposition 4** *Given a diagonal linear recurrent layer augmented with sliding window attention with sliding window of size 3, followed by a MLP layer, and tokens $\boldsymbol{s}_{2j} = \boldsymbol{x}_j$ and $\boldsymbol{s}_{2j+1} = \boldsymbol{y}_j$, for $j = 1, \ldots, N$, drawn from a non-linear model, one can construct recurrent matrix $\boldsymbol{A}(\boldsymbol{s}_j)$, input $\boldsymbol{B}(\boldsymbol{s}_j)$ and output matrix $\boldsymbol{U}(\boldsymbol{s}_j)$ such that each recurrent step for every token $\boldsymbol{s}_j$ produces $\hat{\boldsymbol{y}}_{j+1} = -(\Delta \boldsymbol{W})^T \boldsymbol{x}_{j+1}$ as output, where $\Delta \boldsymbol{W}$ is one step of gradient descent, i.e. $\Delta \boldsymbol{W} = \eta \nabla_{\boldsymbol{W}} \mathcal{L}$. The test input $\boldsymbol{x}_{N+1}$ is contained in token $\boldsymbol{s}_{2N+2}$, and produces the test prediction $\hat{\boldsymbol{y}}_{N+1}$.*

Writing this as an SSM exactly as in 12 (with the same $\boldsymbol{\Psi}$)

$$\boldsymbol{z}_t = \boldsymbol{I}\, \boldsymbol{z}_{t-1} + \boldsymbol{\Psi}\, \boldsymbol{c}_t \,,$$

we now need to calculate the output as

$$\hat{y}_{t+1} = -\eta \nabla_{\boldsymbol{w}_0} \mathcal{L}(\mathcal{D}_{1:t}; \boldsymbol{w}_0)^T \boldsymbol{x}_{t+1} \,.$$

The interleaved MLP layer $f$ would need to learn a mapping such that

$$f\left( \sum_{i=1}^{t} \left( \boldsymbol{w}_0^T \boldsymbol{x}_i - y_i \right) \boldsymbol{x}_i \right) = \sum_{i=1}^{t} \left( g(\boldsymbol{w}_0^T \boldsymbol{x}_i) - y_i \right) g'(\boldsymbol{w}_0^T \boldsymbol{x}_i) \boldsymbol{x}_i \,,$$

which it will be able to since it is an universal approximator. This can then be used to calculate the output as

$$o_t = f(\boldsymbol{z}_t)^T \boldsymbol{\Theta} \boldsymbol{c}_t \,.$$

## A.4 Regularisation terms in the loss

For example, if we wished to change the loss function in Eq. 22 to include L2 regression,

$$\mathcal{L}(\mathcal{D}; \boldsymbol{W}) = \frac{1}{2N} \sum_{i=1}^{N} ||\hat{\boldsymbol{y}}_i - \boldsymbol{y}_i||_2^2 + ||\boldsymbol{W}||_2^2 \,,$$

this will change the gradient to be

$$\nabla_{\boldsymbol{w}} \mathcal{L}(\mathcal{D}_{1:t}; \boldsymbol{W}_0) = \frac{1}{t} \sum_{i=1}^{t} \frac{\partial}{\partial \boldsymbol{W}} ||\hat{\boldsymbol{y}}_i - \boldsymbol{y}_i||_2^2 + 2\boldsymbol{W}_0 \,.$$

To construct an SSM that does GD on this loss instead of the one in Eq. 8, all we'd have to do is change the $\Delta W$ in the output to be

$$\Delta_l \boldsymbol{W} = -\frac{\eta}{N} \left( \boldsymbol{W}_{l-1}^T \tilde{\boldsymbol{Z}}_t^{(l)} - \boldsymbol{Z}_t^{(l)} + \boldsymbol{W}_{l-1} \right) \,.$$

## A.5 Experimental details of 1-d regression

The dataset construction and the model evaluation methods are same as that used in von Oswald et al. (2023). Each task (context) $\tau$ consists of in-context training data $D_\tau = \{(x_{\tau,i}, y_{\tau,i})\}_{i=1}^{N}$ and test point $(x_{\tau,N+1}, y_{\tau,N+1})$. At every optimization step of the model, we sample the regression parameters $W_\tau \sim \mathcal{N}(0, I)$. We then sample $x_{\tau,i} \sim U(-1, 1)^f$ and construct a scalar target $y_{\tau,i} = W_\tau x_{\tau,i}$, where $f$ is the feature size of the input. To evaluate the model, a set of $10^4$ tasks are sampled and mean squared error is calculated.

To evaluate the proposed model, we conducted experiments using a constructed token dataset with an input feature size of 10. The model architecture is a single-layer, state space model (SSM) with a hidden dimension of 20. Initialization was performed with 2 blocks and an SSM latent size of 10. No activation function was used during training.

The training process spanned a maximum of 300,000 epochs with a batch size of 64. A cosine annealing schedule and linear warmup were utilized for the learning rate, beginning with an initial SSM learning rate of $1 \times 10^{-4}$ for optimizing the SSM parameters using the AdamW optimizer. The global learning rate for the remaining parameters was set to $2 \times 10^{-4}$, and these parameters were also optimized using AdamW. A weight decay of 0.05 was applied to regularize the model and prevent overfitting.

## A.6 Experimental details of N-d regression

The setup is similar to 1-d regression, but we sample $n_o$ different regression parameters $W_\tau^k \sim \mathcal{N}(0, I)$, where $n_o$ is the dimension of vector $y_{\tau,i}$ and $1 <= k <= n_o$. We then construct target $y_{\tau,i}^k = W_\tau^k x_{\tau,i}$ for all $k$. For all our experiments, we choose $n_o = f$. We have evaluated the model based on all the experiments that are used for the 1-D regression evaluation.

To have the model prediction equivalent to one step gradient descent, we train the single layer GD-SSM using Adam optimizer, with an initial learning rate of 0.0001 for recurrent parameters with cosine annealing. For all the other parameters we double the learning rate that is used for recurrent parameters. In all our experiments, each optimization step contains 64 tasks.

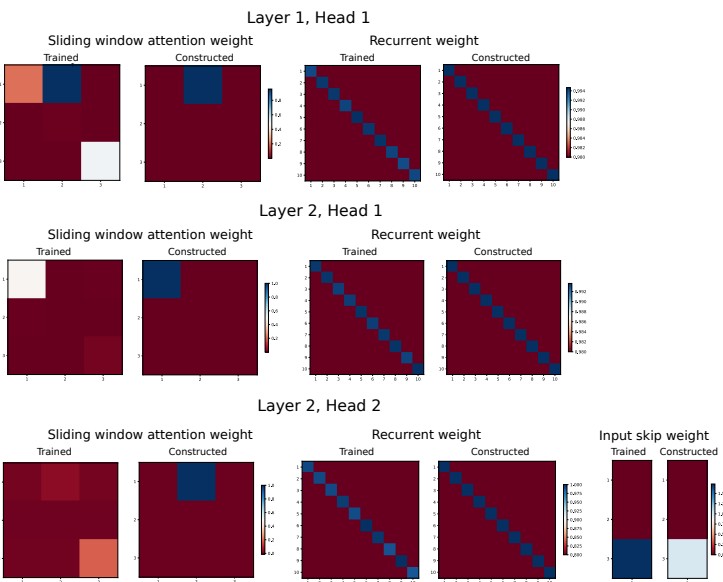

Figure 5: Visualisation of the trained parameters for two layer GD-SSM on linear regression task.

## A.7 Experimental details of model comparison

Table 1 details the model configurations shown in Figure 4C. The base Griffin and Mamba models were both trained using the Adam optimizer with a learning rate of 0.0001, matching S5's configuration. For the enhanced versions, Mamba+Linear was configured with 4 layers and 128 hidden states, trained using the AdamW optimizer with a learning rate of 0.0001, and evaluated on both N-D and 1-D settings. Similarly, Griffin+Linear was structured with 4 layers, utilizing an LRU width of 128, and 4 Multi-Query Attention (MQA) heads, trained with the Adam optimizer at a learning rate of 0.0001 in 1-D setting and 0.0002 in 2-D setting.

The architectural comparison in Table 1 highlights key differences between models. Griffin combines two RG-LRU modules with one MQA module, while Mamba utilizes a more straightforward structure with its basic building blocks. Our experiments show that Griffin achieves its best performance (loss: 0.4114) with 2 heads and 2 layers, with additional parameter scaling showing minimal improvements. Mamba maintains performance across different configurations, with losses ranging from 0.4126 to 0.5388.

In subsequent experiments, we explored the effect of introducing Linear Projection into the Mamba architecture. From the comparison between Figure 6 and Figure 4C, it can be clearly seen that the combination of Mamba+Linear Projection brings significant performance improvement and achieves excellent experimental results.

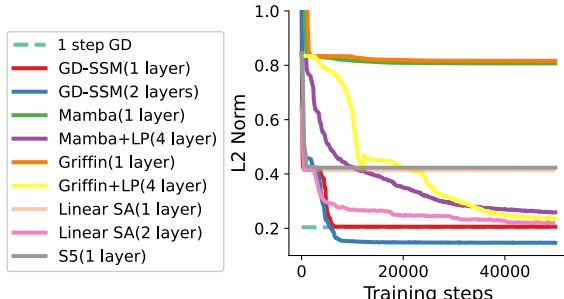

Figure 6: Comparison with other models on N-D regression. The GD-SSM model was evaluated with both 1-layer and 2-layer configurations, and the S5, Mamba, and Griffin models were included for comparison. Linear SA refers to linear self attention models, with both 1-layer and 2-layer variants tested to evaluate their performance.

| Griffin Model | | | | |
|---|---|---|---|---|
| head | 1 | 2 | 2 | 5 |
| layer | 1 | 1 | 2 | 1 |
| loss | 0.7490 | 0.5665 | 0.4114 | 0.6047 |
| Mamba Model | | | | |
| Model dimension | 32 | 64 | 64 | 128 |
| layer | 1 | 1 | 2 | 4 |
| loss | 0.5388 | 0.4126 | 0.4179 | 0.4157 |
| S5 Model | | | | |
| layer | 1 | 2 | 4 | 6 |
| loss | 0.426 | 0.426 | 0.426 | 0.426 |

Table 1: Comparison of Griffin, Mamba and S5 Models with different layers/heads

We also compare the performance of S5 model on the N-dimensional linear regression task with different number of layers. For each S5 model, we use model size and state size of 32, and an input and output projection to transform the input data to the higher dimensional model size and vice-versa. We also experiment with different model size and state size from a set of [10,64,128,256], but do not observe any improvement in the performance. For training, Adam optimizer with a learning rate of 0.0001 is used. We use zero-order hold method to discretize the state space system.

### A.8 EXPERIMENTAL DETAILS OF MULTI-STEP AND NON-LINEAR REGRESSION

To emulate the multi-step regression task, we train the multi-layer GD-SSM architecture, without applying any non-linearity between the layers. In our experiments, we use two layer GD-SSM architecture. We use the same hyper-parameters that are used for the single layer GD-SSM. We compare the validation loss of the regression task between the learned model and the model based on construction. We observe the performance of two layer GD-SSM better than the GD-SSM initialised with the parameters from our construction, so we train the GD-SSM model initialised with the parameters from our construction 1000 steps to match the performance of a trained GD-SSM.

For non-linear regression tasks, we use the experiments from Finn et al. (2017) and follow the data construction of von Oswald et al. (2023). To the output of the linear GD-SSM layer, we add a non-linear function, weighted sigmoid gated unit (Tanaka, 2020), as used in (Smith et al., 2022). For all the non-linear regression tasks, an input embedding layer is used in addition to the model architecture used for linear regression tasks. The hyper-parameters are the same that is used for all the previous tasks, and the trained model loss is compared with a model with GD-SSM layer/layers based on gradient descent construction and trained non-linearity layer/layers. Similar to multi-step linear regression, we observe the GD-SSM performance better than the GD-SSM initialised with the parameters from our construction.

## A.9 ABLATION STUDY: 1-D REGRESSION

This experiment analyzed the impact of the input matrix and output matrix construction on model performance by performing an ablation study on the 1-D GD-SSM model. The experimental results show that when both components exist at the same time, the model loss is the lowest (0.209); while removing either component or both components at the same time will increase the loss to 0.426. This shows that the construction of the input and output matrices is interdependent and needs to exist at the same time to work. Keeping only one of them will not improve the model performance.

| Model | Input Construction | Output Construction | Loss |
|---|---|---|---|
| GD-SSM | ✓ | ✓ | 0.209 |
| GD-SSM | ✗ | ✓ | 0.426 |
| GD-SSM | ✓ | ✗ | 0.426 |
| GD-SSM | ✗ | ✗ | 0.426 |

Table 2: Ablation test on 1-D GD-SSM

## A.10 ABLATION STUDY: N-D REGRESSION

The two important features we propose for emulating gradient descent in our GD-SSM model architecture are the sliding window attention and the multiplicative interaction between the hidden state of the SSM and input. In this ablation study, we compare our proposed one layer GD-SSM model performance with its variants where the sliding window attention and output skip connection are turned off. The model performance shows that both these features are crucial for the model in emulating a single step gradient descent update.

| Model | Sliding window attention | Multiplicative output skip-connection | Loss |
|---|---|---|---|
| GD-SSM | ✓ | ✓ | 0.206 |
| GD-SSM | ✗ | ✓ | 0.41 |
| GD-SSM | ✓ | ✗ | 0.41 |
| GD-SSM | ✗ | ✗ | 0.41 |

Table 3: A comparison of our proposed model GD-SSM with its variants where two major features sliding window attention and output skip connection are turned off.

