# OpenReview forum: "State-space models can learn in-context by gradient descent"
_ICLR.cc/2025/Conference — Submitted to ICLR 2025_

### Official Review · Reviewer_e4sc · 2024-10-17

**Soundness:** 3
**Presentation:** 2
**Contribution:** 1
**Rating:** 3
**Confidence:** 5

**Summary:**

This paper imitates Transformers learn in-context by gradient descent(https://arxiv.org/abs/2212.07677). This paper proves that a single structured state-space model layer with local self-attention can reproduce the outputs of an implicit linear model with least square loss. (The task considered is not general)

**Strengths:**

1. The paper demonstrates that the state-space model with LSA can learn in-context learning tasks on linear and nonlinear regression problems. While I do not dispute the claim that state-space models can achieve in-context learning via gradient descent, my concern lies in whether the specific modification introduced warrants the detailed calculations provided. The conclusions, as presented, seem to offer limited insights into how this work might advance research on improving in-context learning capabilities. A clearer connection to meaningful improvements in this area would significantly enhance the paper's contribution.
2. The experiments compare the state-space model with Griffin and Linear Transformer, but they are restricted to shallow architectures of 1 or 2 layers. This setup is inconsistent with typical in-context learning scenarios, where models need to be sufficiently large for emergent phenomena and meaningful in-context learning capabilities to surface. The current experiments do not effectively capture these dynamics, making it difficult to observe such phenomena in shallow sequence models. Expanding the experiments to include deeper architectures would provide a more realistic assessment of in-context learning in state-space models.

**Weaknesses:**

1. The paper addresses a theoretical statement about the transformer model, specifically that it can learn in-context through gradient descent. However, this concept is already well-known and documented. In theoretical research, it is widely recognized that transformers, convolutional models, and recurrent models, such as state-space models, are universal approximators and are capable of learning continuous target relationships. Therefore, demonstrating the same for state-space models does not appear to offer significant theoretical advancement. If the authors wish to underscore the importance of this work, I would recommend showing that previous work in approximation theory does not extend to the in-context learning case. Without this distinction, the contribution seems to fall within a subset of known results that hold limited value for theoretical study.
2. The notion of in-context learning, as presented, lacks practical interest. Simply stating that a model "can learn" through in-context learning is insufficient, as the same argument could be made for various methods, including Newton's or quasi-Newton's methods. There is no compelling reason for practitioners to assume that, when state-space models engage in in-context learning, the behavior in terms of loss convergence or asymptotic rates would align with that of gradient descent. Clarifying this distinction would strengthen the paper’s contribution. Could you provide empirical comparisons of convergence rates or asymptotic behavior between your method and alternatives like Newton's or quasi-Newton's methods.
3. The paper introduces a state-space model with local self-attention, which is not a commonly adopted approach in practice. It would be more beneficial to align the framework with models that are widely used in real-world applications. A method akin to the linear transformer might be more appropriate and could provide a better point of reference for practical utility.

**Questions:**

1. See Weakness.
2. The behavior of GD and 1-D GD-SSM appears different. Could the authors provide an explanation for this discrepancy? Clarifying this would help the readers better understand the distinctions in learning dynamics between these approaches.
3. Figure 3: The font size in Figure 3 is too small and could be increased for readability.

---

> ### Author Response · Authors · 2024-11-23
>
> > 1. The paper addresses a theoretical statement about the transformer model, specifically that it can learn in-context through gradient descent. However, this concept is already well-known and documented. In theoretical research, it is widely recognized that transformers, convolutional models, and recurrent models, such as state-space models, are universal approximators and are capable of learning continuous target relationships. Therefore, demonstrating the same for state-space models does not appear to offer significant theoretical advancement. If the authors wish to underscore the importance of this work, I would recommend showing that previous work in approximation theory does not extend to the in-context learning case. Without this distinction, the contribution seems to fall within a subset of known results that hold limited value for theoretical study.
>
> The reviewer mentions that the ability of transformers and other universal models to perform ICL is well-documented, and thus demonstrating the same for SSMs might not constitute a significant theoretical advancement. We respectfully clarify that our work provides a mechanistic explanation specifically tied to gradient descent dynamics in SSMs, which distinguishes it from prior general approximation results. While it is true that many architectures can approximate target functions, we focus on elucidating how SSMs can explicitly emulate gradient-based learning mechanisms. Our goal is not to provide a general approximation result but a concrete and direct construction to show that ICL can be implement using GD in SSMs.
>
> > 2. The notion of in-context learning, as presented, lacks practical interest. Simply stating that a model "can learn" through in-context learning is insufficient, as the same argument could be made for various methods, including Newton's or quasi-Newton's methods. There is no compelling reason for practitioners to assume that, when state-space models engage in in-context learning, the behavior in terms of loss convergence or asymptotic rates would align with that of gradient descent. Clarifying this distinction would strengthen the paper’s contribution. Could you provide empirical comparisons of convergence rates or asymptotic behavior between your method and alternatives like Newton's or quasi-Newton's methods.
>
> We are not aware of any specific direct construction showing that SSMs in the general form commonly used can learn using Newton’s or quasi-Newton’s methods. We do in fact demonstrate in Figs 3 & 4 that SSMs trained from a random initialisation do exhibit convergence to the same values as would GD. We plan to look into the similar empirical comparisons for Newton’s and quasi-Newton’s methods in the future work.
>
> > 3. The paper introduces a state-space model with local self-attention, which is not a commonly adopted approach in practice. It would be more beneficial to align the framework with models that are widely used in real-world applications. A method akin to the linear transformer might be more appropriate and could provide a better point of reference for practical utility.
>
> Local self attention is more commonly referred to as sliding window attention in literature (we have updated our paper to reflect this). It is in fact used in major high-performance models such as Griffin [1] and Samba [2].
>
> [1] S. De et al., ‘Griffin: Mixing Gated Linear Recurrences with Local Attention for Efficient Language Models’, Feb. 29, 2024, arXiv:2402.19427. doi: 10.48550/arXiv.2402.19427.
>
> [2] L. Ren, Y. Liu, Y. Lu, Y. Shen, C. Liang, and W. Chen, ‘Samba: Simple Hybrid State Space Models for Efficient Unlimited Context Language Modeling’, Jun. 11, 2024, arXiv:2406.07522. doi: 10.48550/arXiv.2406.07522.

---

> > ### Author Response · Authors · 2024-11-23
> > **responses to questions**
> >
> > > 2. The behavior of GD and 1-D GD-SSM appears different. Could the authors provide an explanation for this discrepancy? Clarifying this would help the readers better understand the distinctions in learning dynamics between these approaches.
> >
> > Fig 2A does not show the learning dynamics of GD but rather the final result obtained by one step of GD. We have updated the figure to make this clearer. The GD-SSM does have a learning curve because we start from a random initialisation of an SSM with the inductive biases we propose and training it on the ICL task for linear regression.
> >
> > > 3. Figure 3: The font size in Figure 3 is too small and could be increased for readability.
> >
> > Thank you, we have fixed this now.

---

> > > ### Comment · Reviewer_e4sc · 2024-11-27
> > >
> > > Thank you for providing the clarifications and additional details. After reviewing them, I have decided to maintain my score. I look forward to future versions that provide a deeper understanding of the relationship between the dynamics of ICL and GD/Newton/Quasi-Newton methods.

---

### Official Review · Reviewer_AvcW · 2024-10-17

**Soundness:** 4
**Presentation:** 3
**Contribution:** 1
**Rating:** 3
**Confidence:** 4

**Summary:**

In-context learning (ICL) is one of the surprising capabilities of LLMs at scale. Seminal results have shown that Transformer-based architectures have this ability and more recent ones confirmed that SSMs also do. This work studies shows that SSMs can implement ICL by gradient descent. They provide a constructive proof showing that 1 layer can implement one GD step and confirm empirically on toy tasks that the networks find this solution.

**Strengths:**

The paper is overall well written and easy to follow. The theoretical part is sound and experiments convincingly demonstrate the paper's claims in toy settings.

**Weaknesses:**

I am concerned by the novelty of this paper. [Zucchet et al. 2023](https://arxiv.org/abs/2309.01775) show that 1 SSM layer can implement any linear self-attention layer. This results implies that any ICL algorithm LSA can implement, an SSM can. This holds for 1 GD step studied in this paper, but also for any other algorithm the community has been studying over the last few years. Additionally, this paper also have very similar experiments to the ones presented here.

**Questions:**

To the best of my understanding and as mentioned above, the results presented here are a subset of the results presented in Zucchet et al. 2023. Can the authors compare their work to that paper and highlight what their insights are?

---

> ### Author Response · Authors · 2024-11-23
>
> We thank the reviewer for recognizing the soundness and clarity of the paper and the empirical evidence supporting the papers claims in small-scale settings.
>
> We also thank the reviewer for bringing up a very closely related paper in Zucchet et al. 2023. However, there are critical differences between their paper and ours as we explain below. To summarize, ours is a more explicit, direct and parsimonious  construction showing that SSMs can do ICL by GD while theirs is very indirect.
>
> In detail:
> In Zucchet et al., they show the equivalence of gated linear recurrent networks and linear self-attention. It is true that linear self-attention has been shown to be able to do ICL by GD [1], and so, indirectly, they show that gated linear recurrent networks can perform ICL by GD.  In our case, we consider the general formulation of SSMs and show a correspondence directly between them and ICL via GD, leading to a simpler and more understandable construction.
>
> Our construction is more parsimonious and efficient due to the following:
> 1.  a model using linear attention model requires $3f^2$ input parameters for the separate embeddings for query, key and value, whereas in our case we only need $f^2$ input parameters since we have only one embedding.
> 2. To train the parameters of a linear-regression model with $f \times f$ parameters the GD-SSM requires $f^2$ recurrent neurons with $f^2$ recurrent parameters since the recurrent matrix is diagonal. Whereas a linear self-attention as constructed in [1] requires $12f^2$ parameters in the linear self-attention.
> 3. The RNN construction in Zucchet et al. uses $O(f^4)$ neurons to represent the $3f^2$ parameters in the linear self-attention layer, further increasing the size and redundancy of the model.
>
> Our construction distils out the exact inductive bias required for ICL. Moreover, the that the simple form of linear transformers used in [1] still lag behind state-space models in task performance.
>
> While the end result is the same, the method used to achieve this is completely different — ours is more direct.
>
> We have added a discussion of this to our paper.
>
> [1] J. von Oswald et al., ‘Transformers Learn In-Context by Gradient Descent’, in Proceedings of the 40th International Conference on Machine Learning, PMLR, Jul. 2023, pp. 35151–35174. Accessed: Apr. 16, 2024. [Online]. Available: https://proceedings.mlr.press/v202/von-oswald23a.html

---

> > ### Comment · Reviewer_AvcW · 2024-11-27
> >
> > Dear authors,
> >
> > Thank you for clarifying the link with Zucchet et al. 2023 [1]. While I agree with the fact that your construction is more direct, as it directly tackles GD whereas [1] is a more general results, I don't think this does not bring any new conceptual insight nor that it is a completely different method. Additionally, the comparison with [1] is a bit misleading in your point 3 is misleading: [1] uses $O(f^2)$ neurons and not $O(f^4)$.
> >
> > For this reason, I decide to keep my score as it is.

---

### Official Review · Reviewer_aDbU · 2024-10-29

**Soundness:** 2
**Presentation:** 3
**Contribution:** 2
**Rating:** 5
**Confidence:** 3

**Summary:**

This paper investigates how state-space models (SSMs) can perform in-context learning through gradient descent. The authors provide both theoretical and empirical evidence that SSMs augmented with local self-attention can emulate gradient descent on implicit regression models. Their key insight is that the diagonal linear recurrent layer in SSMs can act as a gradient accumulator.

**Strengths:**

- To my knowledge, the insight viewing SSMs as a gradient accumulator allowing them to emulate gradient descent on in-context learning tasks is novel and the combination with local self-attention for preprocessing is interesting
- The mathematical theory appears to be sound
- The presentation of the material and step by step walk through of the theory from simple cases to more complex is clear and helpful
- The theoretical findings potentially point to a mechanistic understanding of architectural requirements (for SSMs or other models) that enable types of in-context learning

**Weaknesses:**

- While the paper does a good job of explaining the theory and formulation of GD-SSM and empirically validating GD-SSM on regression tasks, I didn't feel like it shed that much insight into the currently used SSM variants used in practice.
   - Note that line 488 says: "These findings not only explain the success of recent SSM variants in in-context learning tasks but
also provide valuable insights for the design of future sequence models."
   -  Note that Lines 052-053 say, referring to the modern SSM variants : "Which features of these successful models contribute to in-context learning, as opposed to earlier variants? Using a constructive approach, we pinpoint input-dependent input and output processing, as the key features required for in-context learning".
   - But I do not think the current version of the paper supports either of the claim that this paper sheds light on these questions
   - The approach constructed seems to be very different from those used in practice. In addition, the methods commonly used in practice do not appear to do well on the regression ICL tasks in this paper. So it is unclear what I should take away from this related to prior SSM methods?
   - I think the empirical results were an opportunity to provide insight here, but didn't seem to fully achieve this. Please see my questions below which may clarify this for me.

- Related to the above, the experimental section is light on architectural details, making it hard to determine what exactly is being compared empirically and what conclusions can be drawn. Please include more experimental details including the architectures and hyperparameters.

- The paper is often unclear on the terminology of local self-attention and local linear self-attention. The formulation in Section 3.2 appears to only require local linear self-attention, yet other times in the paper local self-attention is used. In the related works section the two are contrasted. I would recommend being very explicit and consistent on this point as the two are very different.

- The paper is limited to regression style in-context learning. This is interesting and amenable to theoretical analysis, but also limits the impact of the investigation. See https://arxiv.org/abs/2401.12973 and https://arxiv.org/abs/2402.04248 for other papers that have empirically investigated other types of in-context learning in different architectures.

**Questions:**

1. Can you explicitly define GD-SSM? Perhaps even provide pseudo-code? What are its learnable parameters? It is defined in 283 after the fact, but I think a more explicit definition would be helpful.

Questions related to experiments and first two bullets in Weaknesses section above:

2. How do the results in this paper explain the success of recent SSM variants (as claimed in lines 488)? Line 052-053 say : "Which features of these successful models contribute to in-context learning, as opposed to earlier variants? Using a constructive approach, we pinpoint input-dependent input and output processing, as the key features required for in-context learning". I hoped the paper would provide insight into this question. However, it seems that the constructed GD-SSM is quite different from the currently used SSM architectures (e.g. Griffin, Mamba), and performs much better empirically in the experiments. Meanwhile the currently used SSM architectures do not seem to perform regression in-context learning (at least for one layer). So how do the results in this paper answer (or point to answering) the first question in line 052 or support the claim in line 488? This is my main question regarding this paper. I list some additional questions below that are an attempt to clarify some of the presented empirical results below.

3. Why does Griffin do so poorly compared to linear transformers and S5? Shouldn't Griffin be a mix of sliding window attention and SSM (the RG-LRU), making it similar to the GD-SSM formulation? Or is the model referred to as Griffin just RG-LRU?

4. Why does the time-invariant S5 appear to consistently outperform the input-dependent Mamba and Griffin models (even though all fail to converge)? I would have expected the models with input-dependent dynamics to perform ICL better.  Is this difference at all interesting?

5. Does it benefit the other architectures (1 layer linear attention, S5, Griffin, Mamba) to also provide them with the local self-attention preprocessing? Shouldn't they then be able to potentially learn the GD-SSM model if provided this? If not, why not?

6. Can 2 layers of the SSMs solve the task? Note that combining the local self attention with the diagonal SSM is a combination of 2 sequence processors.


Other questions and comments:

7. Where is the 1-step GD in Figure A?

8. In Figure 4B, the 1 or 2 layer GD-SSM formulation always seems to achieve lower loss than the corresponding number of steps GD model. Why is this? What happens if we compare more layers and more steps, e.g. 5 or 10 steps/layers?

9. Note that the color scheme in Figure 4C etc is hard to read and tell which line corresponds to which model.

10. Note that LSA is first introduced in equation 2, but only later defined in line 119.

---

> ### Author Response · Authors · 2024-11-23
> **thank you for a constructive review**
>
> We thank the reviewer for their constructive points, and appreciate the reviewer recognizing the novelty of our perspective on SSMs as gradient accumulators, the soundness of our mathematical theory, and the clarity of our step-by-step theoretical exposition.
>
> Responses to weaknesses:
>
> > While the paper does a good job of explaining the theory and formulation of GD-SSM and empirically validating GD-SSM on regression tasks, I didn't feel like it shed that much insight into the currently used SSM variants used in practice.
>
> We agree that the constructed approach doesn’t exactly match existing methods, although the use of sliding window attention is becoming more common on highly-performant models [1,2] and 2-D state and input-dependent output computation is pretty standard through gating [3].
> To explore why Griffin and Mamba do so badly, we explored their performance a bit more (see Table 1 in appendix). We found that adding additional layers (our earlier experiments were with 1-layer), increasing the model dimension, and increasing the number of heads improves performance . We hypothesize that this is due to the fact that the architecture doesn’t exactly match GD, and additional layers are required to play the role of the sliding window attention to perform time-mixing (access adjacent elements in the sequence).
>
> Our construction distils out the exact inductive bias required for ICL.
>
> > Related to the above, the experimental section is light on architectural details, making it hard to determine what exactly is being compared empirically and what conclusions can be drawn. Please include more experimental details including the architectures and hyperparameters.
>
> We’ve added more details in the Appendix about the details of the experiments and architectural details. Please let us know if any specific aspects still remain unclear.
>
> > The paper is often unclear on the terminology of local self-attention and local linear self-attention. The formulation in Section 3.2 appears to only require local linear self-attention, yet other times in the paper local self-attention is used. In the related works section the two are contrasted. I would recommend being very explicit and consistent on this point as the two are very different.
>
> We agree with the reviewer, and have changed all references to “Local Self-attention” into "Sliding Window Attention” which is more explicit.
>
> > The paper is limited to regression style in-context learning. This is interesting and amenable to theoretical analysis, but also limits the impact of the investigation. See https://arxiv.org/abs/2401.12973 and https://arxiv.org/abs/2402.04248 for other papers that have empirically investigated other types of in-context learning in different architectures.
>
> We agree with the reviewer, and plan to look at other ICL tasks in future work.
>
> [1] S. De et al., ‘Griffin: Mixing Gated Linear Recurrences with Local Attention for Efficient Language Models’, Feb. 29, 2024, arXiv:2402.19427. doi: 10.48550/arXiv.2402.19427.
>
> [2] L. Ren, Y. Liu, Y. Lu, Y. Shen, C. Liang, and W. Chen, ‘Samba: Simple Hybrid State Space Models for Efficient Unlimited Context Language Modeling’, Jun. 11, 2024, arXiv:2406.07522. doi: 10.48550/arXiv.2406.07522.
>
> [3] A. Gu and T. Dao, ‘Mamba: Linear-Time Sequence Modeling with Selective State Spaces’, Dec. 01, 2023, arXiv: arXiv:2312.00752.
>
> [4] T. Dao and A. Gu, ‘Transformers are SSMs: Generalized Models and Efficient Algorithms Through Structured State Space Duality’, in Proceedings of the 41st International Conference on Machine Learning, PMLR, Jul. 2024, pp. 10041–10071.
>
> [5] J. Park et al., ‘Can Mamba Learn How to Learn? A Comparative Study on In-Context Learning Tasks’, Feb. 06, 2024, arXiv: arXiv:2402.04248.

---

> > ### Author Response · Authors · 2024-11-23
> > **responses to questions**
> >
> > > 1. Can you explicitly define GD-SSM? Perhaps even provide pseudo-code? What are its learnable parameters? It is defined in 283 after the fact, but I think a more explicit definition would be helpful.
> >
> > We have now mentioned the term GD-SSM earlier in 061 and added references to it close to l.285 near the equations that define it for the general case. We have also described which parameters of the model are learnable over there.
> >
> > > 2. How do the results in this paper explain the success of recent SSM variants (as claimed in lines 488)? Line 052-053 say : "Which features of these successful models contribute to in-context learning, as opposed to earlier variants? Using a constructive approach, we pinpoint input-dependent input and output processing, as the key features required for in-context learning". I hoped the paper would provide insight into this question. However, it seems that the constructed GD-SSM is quite different from the currently used SSM architectures (e.g. Griffin, Mamba), and performs much better empirically in the experiments. Meanwhile the currently used SSM architectures do not seem to perform regression in-context learning (at least for one layer). So how do the results in this paper answer (or point to answering) the first question in line 052 or support the claim in line 488? This is my main question regarding this paper. I list some additional questions below that are an attempt to clarify some of the presented empirical results below.
> >
> > Please see the explanation we have provided above on this point in the weaknesses.
> >
> > > 3. Why does Griffin do so poorly compared to linear transformers and S5? Shouldn't Griffin be a mix of sliding window attention and SSM (the RG-LRU), making it similar to the GD-SSM formulation? Or is the model referred to as Griffin just RG-LRU?
> >
> > While Griffin does have sliding window attention, we hypothesize that it’s poor performance might be due to the specific form of recurrence it has which doesn’t match GD. Adding additional layers and heads seems to improve performance — see new results added in Table 1 in Appendix.
> >
> > > 4. Why does the time-invariant S5 appear to consistently outperform the input-dependent Mamba and Griffin models (even though all fail to converge)? I would have expected the models with input-dependent dynamics to perform ICL better. Is this difference at all interesting?
> >
> > From our analysis, it doesn’t seem like input-dependent dynamics itself has a major role to play for ICL. Note that input-dependent recurrence would correspond to online GD rather than mini-batch GD, which is inherently noisier. It is certainly a very interesting aspect, and we plan to explore it further to understand how this construction compares to how existing models do ICL in future work.
> >
> > > 5. Does it benefit the other architectures (1 layer linear attention, S5, Griffin, Mamba) to also provide them with the local self-attention preprocessing? Shouldn't they then be able to potentially learn the GD-SSM model if provided this? If not, why not?
> >
> > We do expect that adding SWA to existing architectures would enable them to learn GD-SSM as long as they also have multiplicative input-dependent output processing (through gating potentially) and 2-D state or equivalent.
> >
> > > 6. Can 2 layers of the SSMs solve the task? Note that combining the local self attention with the diagonal SSM is a combination of 2 sequence processors.
> >
> > Yes, we expect so. Our current results already demonstrates that a 2-layer linear self-attention can solve the linear regression task — See Fig. 4C and Fig. 6. We expect it would be similar for other SSMs (as long as the output processing is appropriate).
> >
> > > 7. Where is the 1-step GD in Figure A?
> >
> > Thank you for pointing this out. The 1-step GD loss is the same as in Fig 2A. We’ve added it now to Fig 4A.
> >
> > > 8. In Figure 4B, the 1 or 2 layer GD-SSM formulation always seems to achieve lower loss than the corresponding number of steps GD model. Why is this? What happens if we compare more layers and more steps, e.g. 5 or 10 steps/layers?
> >
> > We suspect that in the case of non-linear regression, GD-SSM learns a strategy that is presumably better than pure GD. But it’s not clear what this strategy might be.
> >
> > > 9. Note that the colour scheme in Figure 4C etc is hard to read and tell which line corresponds to which model.
> > > 10. Note that LSA is first introduced in equation 2, but only later defined in line 119.
> >
> > Thank you for pointing these out. We have fixed it.

---

> > > ### Comment · Reviewer_aDbU · 2024-11-25
> > >
> > > I thank the authors' for their response. I will take it into account when discussing with the other reviewers during the reviewer discussion phase.

---

### Official Review · Reviewer_pFrz · 2024-11-07

**Soundness:** 3
**Presentation:** 3
**Contribution:** 2
**Rating:** 5
**Confidence:** 5

**Summary:**

The paper investigates the ability of SSMs to perform ICL through gradient descent during the forward pass over recurrent steps. It provides a theoretical analysis of various ICL scenarios, demonstrating that simple SSMs, equipped with input and output-dependent processing, can accurately mimic gradient descent when data is presented as a sequence. This theory offers an explanation for the capacity of modern SSMs to execute ICL and outlines a network circuit capable of handling such tasks. Empirical experiments confirm that the proposed circuit can effectively learn and perform small-scale synthetic tasks in practice.

**Strengths:**

(1) Enhances understanding of inductive biases in SSMs for ICL tasks.

(2) Bridges key domains such as SSMs, ICL, and mechanistic interpretability.

(3) Demonstrates generality by extending results to multi-step cases, multiple layers, and multi-dimensional data.

(4) Empirical analysis shows practical alignment with the theory-based construction in simple cases.

(5) Clarity: While some aspects could be improved (such as adding a figure to illustrate the main ideas in Section 3.1), the paper is clear, well-motivated, and easy to follow. It starts with simple cases and provides clear definitions, making it an enjoyable read!

**Weaknesses:**

(1) The authors have **overlooked important related work in this domain**. For example, [1] presented at NeurIPS 2016, demonstrates that **RNNs can perform gradient descent during the forward pass**. Additionally, Section 3.1 in the current paper shares several similarities with Section 2 of [1]. To be clear, this is not an accusation of plagiarism, but rather an indication of missing key references. While there is a difference in scope (RNNs versus SSMs), this oversight reduces the originality of contribution #1. I kindly ask the authors to specify the principal differences between the approach taken by [1] and the approach used by SSMs in their work to better highlight the novel contributions.

(2) **Overlooks simple alternative approaches:** While the theoretical analysis is accurate, the authors overlook significant alternative approaches. References [2-4] demonstrate the connection between S6 layers and attention, showing that S6 can be more expressive than attention without the softmax. Additionally, various ICL studies for transformers omit the softmax (see [5-6] as an example), allowing direct conclusions that could extend to SSMs. **Given the extensive exploration of ICL capabilities in transformers, discussions on the ICL potential of SSMs should consider this reduction approach**. I recommend that the authors evaluate whether this approach could yield additional theoretical developments to strengthen their analysis. Moreover, I suggest that the authors explicitly state the advantages of their approach compared to the proposed alternatives. For instance, Section 3 introduces specific circuits that cannot be achieved through simple reductions.

(3) **Understanding ICL in SSM variants can be enhanced** by examining their sub-components. Previous research indicates that S6 layers exhibit significantly better ICL capabilities than earlier SSMs [7]. However, while the authors highlight input and output-dependent processing as crucial features, they do not empirically ablate these features across various ICL tasks, nor do they provide a detailed theoretical analysis to substantiate this claim explicitly. I recommend adding a subsection that explores these aspects in depth. It is also important to note that input- and output-dependent processing can be implemented through various gating mechanisms. Hence, this claim could be considered somewhat ambiguous, as gated state-space models have been previously studied without demonstrating the same level of ICL capabilities as models like Mamba / S6.


(4) **The claims regarding GD-SSM** (“Our construction, which we call GD-SSM, is not restricted to in-context learning tasks and performs well on general-purpose prediction problems”) do not hold, and much **more empirical analysis is required to justify** them.

___

[1] Learning to learn by gradient descent by gradient descent. Andrychowicz et al.

[2] The Hidden Attention of Mamba Models. Ali et al.

[3] Transformers are SSMs: Generalized Models and Efficient Algorithms Through Structured State Space Duality. Dao et al.

[4] Understanding the differences in Foundation Models: Attention, State Space Models, and Recurrent Neural Networks. Sieber et al.

[5] Transformers Learn In-Context by Gradient Descent. Oswald et al. (see section 2)

[6] Why Can GPT Learn In-Context? Language Models Implicitly Perform Gradient Descent as Meta-Optimizers. Dai et al. (see section 3.1)

[7] Can mamba learn how to learn? a comparative study on in-context learning tasks. Park et al.

**Questions:**

Please see weaknesses 1-3.

---

> ### Author Response · Authors · 2024-11-23
>
> We thank the reviewer for recognizing the strengths of the paper including its clarity, ability to bridge key domains and helping enhance the understanding of inductive biases for ICL tasks. We respond to listed weaknesses below:
>
> > (1) The authors have overlooked important related work in this domain. For example, [1] presented at NeurIPS 2016, demonstrates that RNNs can perform gradient descent during the forward pass.
>
> We appreciate the reviewer bringing up [1] (Andrychowicz et al., NeurIPS 2016). However, we believe this work is completely unrelated to ours. Reference [1] employs meta-learning to train two levels of LSTMs: one LSTM generating weight updates which are used to update the “lower level” LSTM (see Fig. 3 in [1]). This meta-learning framework fundamentally differs from our work, where we construct a state-space model (SSM) that implicitly performs gradient descent as part of its forward pass dynamics to perform in-context learning. We do not use meta-learning, and we do not use LSTMs producing updates for another LSTM.
>
> > Additionally, Section 3.1 in the current paper shares several similarities with Section 2 of [1]. To be clear, this is not an accusation of plagiarism, but rather an indication of missing key references. While there is a difference in scope (RNNs versus SSMs), this oversight reduces the originality of contribution #1. I kindly ask the authors to specify the principal differences between the approach taken by [1] and the approach used by SSMs in their work to better highlight the novel contributions.
>
> We would appreciate a more detailed explanation of the reviewer’s concern on the similarity between Section 2 of [1] and our Section 3.1. To our understanding [1] does not consider linear regression at all, and in Section 2, formulates a meta-learning loss. Whereas, in our Section 3.1, we start from linear regression and construct an SSM that can perform in-context learning. Our approach provides a mechanistic explanation for how in-context learning can emerge in SSMs without requiring meta-learned updates or external optimization signals.
>
> > (2) Overlooks simple alternative approaches: While the theoretical analysis is accurate, the authors overlook significant alternative approaches. References [2-4] demonstrate the connection between S6 layers and attention, showing that S6 can be more expressive than attention without the softmax. Additionally, various ICL studies for transformers omit the softmax (see [5-6] as an example), allowing direct conclusions that could extend to SSMs. Given the extensive exploration of ICL capabilities in transformers, discussions on the ICL potential of SSMs should consider this reduction approach. I recommend that the authors evaluate whether this approach could yield additional theoretical developments to strengthen their analysis. Moreover, I suggest that the authors explicitly state the advantages of their approach compared to the proposed alternatives. For instance, Section 3 introduces specific circuits that cannot be achieved through simple reductions.
>
> We recognize the relevance of studies [2-6] demonstrating connections between SSMs, attention mechanisms, and gradient descent. While those studies propose **indirect** reduction-based approaches, our GD-SSM construction **explicitly and directly** encodes gradient descent into the recurrent dynamics of SSMs, without relying on architectural simplifications. Therefore, we contend that our approach is simpler.
>
> While it is true that linear self-attention can be written as a recurrent network, our approach demonstrates a more direct connection between state-space models and ICL by GD.  Note that the simple form of linear transformers used in [5] still lag behind state-space models in task performance.  Moreover our construction is more parameter efficient than linear self-attention —  a model using linear attention model requires $3mf$ input parameters for the separate embeddings for query, key and value, whereas in our case we only need $mf$ input parameters since we have only one embedding.
>
> References [2-4] deals with selective state-space models with non-linear input-dependence of the SSM parameters. Our construction suggests that such non-linear Input dependence, while powerful, may not be necessary for in-context learning. The remaining studies are empirical, and don’t show a direct theoretical construction, but are nonetheless interesting to understand if the ICL mechanism in the wild corresponds to the one we propose.
>
> As suggested, we will expand our discussion to explicitly outline the advantages of our approach over reduction-based methods, such as greater interpretability and architectural alignment with currently used SSM and clarify that the unique circuits introduced in Section 3 cannot be achieved through these reductions alone. This addition will provide a clearer context for how GD-SSM complements and extends prior work.

---

> > ### Author Response · Authors · 2024-11-23
> >
> > > (3) Understanding ICL in SSM variants can be enhanced by examining their sub-components. Previous research indicates that S6 layers exhibit significantly better ICL capabilities than earlier SSMs [7]. However, while the authors highlight input and output-dependent processing as crucial features, they do not empirically ablate these features across various ICL tasks, nor do they provide a detailed theoretical analysis to substantiate this claim explicitly. I recommend adding a subsection that explores these aspects in depth. It is also important to note that input- and output-dependent processing can be implemented through various gating mechanisms. Hence, this claim could be considered somewhat ambiguous, as gated state-space models have been previously studied without demonstrating the same level of ICL capabilities as models like Mamba / S6.
> >
> > Our theoretical analysis demonstrates a construction that does show that input-dependent processing both at the inputs and outputs is necessary to achieve ICL. We agree that these can be implemented by various gating mechanisms, but our construction suggests that they need to be of a specific form to achieve ICL. Specifically, the input-dependent input processing needs to be in the form of a linear sliding window attention. To our knowledge, previous gated state-space models do not consider this form. We have also added empirical ablation studies in the Appendix in the updated version of the paper.
> >
> > > (4) The claims regarding GD-SSM (“Our construction, which we call GD-SSM, is not restricted to in-context learning tasks and performs well on general-purpose prediction problems”) do not hold, and much more empirical analysis is required to justify them.
> >
> > We acknowledge that the statement regarding GD-SSM’s applicability to general prediction tasks requires additional empirical evidence, and have updated this in the paper. We plan to look at other sequence modelling tasks including language modelling in our future work.

---

> > > ### Comment · Reviewer_pFrz · 2024-11-26
> > >
> > > Thank you for your clarifications and the added details.
> > >
> > > First, thank you for the clarification regarding W.1. I completely mixed up the references and apologize for the confusion. Your response fully addresses this issue.
> > >
> > > Regarding W.2 (direct vs. indirect approach), I agree that the proposed formulation is more parameter-efficient, direct, and simpler than the indirect approach. However, I believe these advantages should be clearly discussed (in depth) in the paper and demonstrated as significant through theoretical or empirical findings. Without this, in my view, the main contribution feels somewhat limited.
> > >
> > > Additionally, concerns W.3 and W.4 remain unaddressed. Therefore, while I raise my score from 3 to 5, I still believe that this paper should be rejected.

---

### Meta-Review · Area_Chair_DZNL · 2024-12-26

**Metareview:**

The paper proposes that ssms can perform in-context learning through gradient descent during their forward pass. The authors try to provide theoretical and empirical evidence that SSMs augmented with local self-attention can emulate gradient descent on implicit regression models. Their key insight is that the diagonal linear recurrent layer in ssms can act as a gradient accumulator.

positive:
- novel theoretical perspective on ssms as gradient accumulators for in-context learning
- sound mathematical theory

Weaknesses:

- limited practical insights into currently used ssm variants (e.g., Griffin, Mamba)
- restricted to regression-style in-context learning tasks
- unclear connection between theoretical construction and empirical success of modern ssms
- missing related works.
- more empirical support is needed for the claims about GD-SSM's general applicability.

The paper is borderline. While authors addressed some of reviewers' concerns, they remained skeptical overall. I vote for a rejection but encourage the authors to improve the paper based on reviewers' suggestions.

**Additional Comments On Reviewer Discussion:**

The rebuttal period focused on several key points:

- the authors clarified their construction is more direct and parameter-efficient than indirect reduction-based approaches, though reviewer AvcW maintained this didn't provide sufficient new conceptual insights.

- the authors added empirical ablation studies and architectural details in response to concerns about connections to practical architectures. They showed that adding layers and increasing model dimension improved the performance of existing architectures.

- the authors acknowledged the need to tone down claims about GD-SSM's applicability to general prediction tasks.

While the authors provided thorough responses, the fundamental concerns about novelty and practical relevance weren't fully addressed.

---

### Decision · Program_Chairs · 2025-01-22

Reject